# Forest carbon stocks increase with higher dominance of ectomycorrhizal trees in high latitude forests

Guoyong Yan [1], Chunnan Fan[2], Junqiang Zheng[3], Guancheng Liu[1], Jinghua Yu[4], Zhongling Guo[2], Wei Cao[4], Lihua Wang[4], Wenjie Wang[5], Qingfan Meng[2], Junhui Zhang[1], Yan Li[2], Jinping Zheng[2], Xiaoyang Cui[5], Xiaochun Wang[5], Lijian Xu [6], Yan Sun[6], Zhi Zhang[7], Xiao-Tao Lü [4], Ying Zhang[4], Rongjiu Shi[4], Guangyou Hao[4], Yue Feng[4], Jinsheng He [8], Qinggui Wang [1] ✉, Yajuan Xing [1,6] ✉ & Shijie Han [1,3,4] ✉

Understanding the mechanisms controlling forest carbon accumulation is crucial for predicting and mitigating future climate change. Yet, it remains unclear whether the dominance of ectomycorrhizal (EcM) trees influences the carbon accumulation of entire forests. In this study, we analyzed forest inventory data from over 4000 forest plots across Northeast China. We find that EcM tree dominance consistently exerts a positive effect on tree, soil, and forest carbon stocks. Moreover, we observe that these positive effects are more pronounced during unfavorable climate conditions, at lower tree species richness, and during early successional stages. This underscores the potential of increasing the dominance of native EcM tree species not only to enhance carbon stocks but also to bolster resilience against climate change in high-latitude forests. Here we show that forest managers can make informed decisions to optimize carbon accumulation by considering various factors such as mycorrhizal types, climate, successional stages, and species richness.

Forests cover approximately 30% of the land surface and store about 45% of terrestrial carbon (C) stocks[1,2]. Additionally, forests have sequestered as much as 30% of annual global anthropogenic carbon dioxide ($CO_2$) emissions[2,3]. Thus, maintaining and enhancing C stocks in forests has been highlighted as a crucial component of pathways to mitigate anthropogenic climate change[4,5]. Theoretical and experimental studies have underscored that tree functional composition holds the potential to significantly influence forest C sequestration[6–10]. However, the role of plant-associated mycorrhizal fungal symbionts in shaping plant functional variation remains understudied[11,12].

Mycorrhizas represent the most widespread mutualistic symbioses established between plant roots and soil fungi[13]. Globally, mycorrhizas play various roles, including mediating nutrient cycling[14], altering the respiratory activity of free-living soil microorganisms[15], shaping soil C sequestration[16], determining how plants respond to climate change[17], and affecting plant population dynamics and species diversity by improving plant fitness and altering the strength of con-specific negative density dependence[18,19]. The two dominant types of mycorrhizal symbiosis, associated with nearly all tree species, are ectomycorrhizal (EcM) fungi and arbuscular mycorrhizal (AM) fungi, but their forms and functions greatly differ[13]. In AM-dominated stands,

[1]School of Life Sciences, Qufu Normal University, Qufu 273165, China. [2]School of Forestry, Beihua University, Jilin 132013, China. [3]School of Life Sciences, Henan University, Kaifeng 475004, China. [4]Institute of Applied Ecology, Chinese Academy of Sciences, Shenyang 110016, China. [5]School of Forestry, Northeast Forestry University, Harbin 150040, China. [6]College of Modern Agriculture and Ecological Environment, Heilongjiang University, Harbin 150080, China. [7]College of Ecology, Lishui University, Lishui 323000, China. [8]College of Urban and Environmental Sciences, Peking University, 100871 Beijing, China. ✉e-mail: qgwang1970@163.com; xingyajuan@163.com; hansj@iae.ac.cn

an inorganic nutrient economy prevails due to rapid mineralization of plant-derived C and nutrients, whereas in EcM-dominated stands, an organic nutrient economy dominates owing to the slow turnover of plant-derived C[14,20].

Accordingly, EcM-associated trees can acquire significant amounts of nitrogen (N) from soil organic matter (SOM) compared to AM-associated trees[14,21]. Moreover, EcM-associated trees can compete for organic N with other free-living decomposer microbes in the soil, intensifying the N limitation of free-living decomposer activity[22]. This competition reduces the production of enzymes that degrade SOM, thereby slowing the rate of SOM decomposition and increasing soil C stocks[15,23,24]. Additionally, EcM-associated trees tend to produce lower-quality litter chemically, which decomposes more slowly and inhibits nutrient mineralization[25]. Cotrufo et al. [26] also found that low quality plant inputs and EcM associations, along with climate-driven microbial inhibition, together lead to relatively large accumulations of soil carbon stocks. In contrast, most AM trees produce higher-quality litter that decomposes more rapidly and promotes nutrient mineralization[14,25]. This suggests that EcM-dominated forests cycle C and nutrients conservatively, while AM-dominated forests have more "open" C and nutrient cycles. It is therefore reasonable to predict that EcM-dominated forests may have higher C stocks than AM-dominated forests in nitrogen-restricted ecosystems, although this prediction has been insufficiently tested. Despite increasing recognition of the importance of mycorrhizas in forest functioning, critical gaps persist in our understanding of the role of different types of mycorrhizal-dominated forests in modulating forest C stocks.

While numerous studies have evaluated the direction and magnitude of mycorrhizal effects on forest ecosystem functioning within specific environments and levels of species diversity[9,13,27,28], relatively few have quantified how changes in climate, forest development stages (such as successions), and species composition (such as species diversity) regulate these effects. For instance, Luo et al. [11] examined the effects of mycorrhizal strategies on forest productivity across different levels of species richness. They found that the strength of the effect of mixing mycorrhizal strategies (i.e., both EcM and AM trees present) on forest productivity was more pronounced at low levels of tree species diversity than at high levels, suggesting the need to consider species diversity when studying mycorrhizal effects on C stocks in forests. Climate and forest successional stages can significantly alter the forest environment[29,30], affecting factors such as species diversity, soil nutrients, and microbial communities. Yan et al. [31] also found that climate (temperature and precipitation) regulated the effects of mycorrhizal fungi on C stock. However, depending on mycorrhizal fungal type, mycorrhizal colonisation could mitigate the impact of climate change on plant production via mycorrhizal promotion of stress tolerance and mycorrhizal alleviation of nutrient limitation[32]. Therefore, mycorrhizal fungi's effects on C stocks in high-latitude forests may be regulated by climate, successional stages, and species diversity.

In this study, we investigate the relationship between EcM versus AM-associated tree dominance and forest C stocks, and examine whether these relationships are influenced by climate, forest successional stages, and tree species diversity. We test three hypotheses regarding the effects of tree mycorrhizal dominance on forest C stocks (Fig. 1). Firstly, compared to AM-associated trees, EcM-associated trees possess the ability to acquire N from soil organic matter (SOM), compete for organic N with other free-living decomposer microbes, and produce low-quality litter that degrades more slowly[14,15,24]. Thus, we hypothesized that EcM-dominated stands could accelerate N cycling and absorption while slowing down the decomposition rate of litter, ultimately leading to enhanced tree C stocks, soil C stocks and total forest C accumulation (Fig. 1b). Secondly, we hypothesized that the effects of EcM mycorrhizal associations on tree C stocks, soil C stocks and forest C stocks would be weaker in favorable climatic

conditions (temperature and precipitation exhibit minor seasonal fluctuations) than in unfavorable ones (there are significant seasonal fluctuations in temperature and precipitation), in late succession compared to early succession, and in species-rich stands compared to species-poor stands (Fig. 1b). This shift in tree interactions—from competition to facilitation or cooperation—occurs across a spectrum of climatic conditions, ranging from favorable to unfavorable[31], and throughout different successional stages, spanning from early to late stages[33–36]. Moreover, additional dimensions of functional diversity could improve resource partitioning in stands with high species richness, potentially reducing the stimulative effect (such as SOM mining for N) of EcM mycorrhizal associations. Thus, we further hypothesized that the underlying mechanisms of positive EcM mycorrhizal association effects largely depend on changes in tree functional diversity along climate, succession and species richness (Fig. 1b). Specifically, climate, succession and species richness are a control on EcM dominance and tree functional diversity while also directly affecting forest C stocks.

In this work, we make use of comprehensive grid-based forest inventory data from the Science and Technology Basic Resources Survey Special Key Project (STBRSSKP) of the Ministry of Science and Technology (MOST) of China (2015–2019) to address these ecological questions. All tree species in each sampling plot are identified and categorized as either EcM or AM-associated trees, excluding species with other mycorrhizal strategies (e.g., ericoid mycorrhizal or non-mycorrhizal). Tree C stock for each plot is computed by summing the tree C stocks of all individual trees. Soil C stock of each plot is calculated using soil C concentration and bulk density, and forest C stock of each plot are computed by aggregating the tree C stock and soil C stock for each plot. Tree species richness is used as a measure of tree species diversity, while EcM (or AM) tree dominance is calculated by dividing the total basal area of EcM (or AM) tree species by the total basal area of the stand in each plot. Succession serves as a fundamental concept in ecological theory, yet few studies have attempted broad generalizations of mycorrhizal fungi effects on forest C stocks across a range of successional sites[28]. To address this, we divide all plots into three successional stages (including early, middle and late succession) across broad spatial scales, taking into account different successional types (e.g., primary versus secondary succession) and trajectories.

## Results

### Effects of EcM tree dominance on forest C stocks
Consistent with our hypothesis, all components of C stocks, including tree, soil, and forest C stocks, exhibited a significant positive relationship with EcM tree dominance ($p < 0.001$; indicated by the solid line in Fig. 2; see also Supplementary Fig. S1 for an opposite pattern with AM tree dominance). Additionally, EcM tree dominance emerged as a significant predictor, explaining a substantial amount of variation in tree, soil, and forest C stocks (the sum of tree C stocks + soil C stocks) (see Supplementary Table S1–S2 and Fig. S2; $p < 0.001$ for the EcM tree dominance). The results from fitted random forest models further underscored the importance of the EcM tree dominance as a predictor of C stocks in forests (Fig. S2). The results also revealed that species richness primarily accounted for the variation in tree C stocks, while climate and topographical factors predominantly explained the variation in soil and forest C stocks (see Supplementary Table S1).

### Regulating role of climate, succession and species richness
The EcM tree dominance exhibited significant interactions with climate, successional stages (SUS), and species richness (SR) in influencing tree C stocks. Additionally, it interacted with climate and species richness (SR) in affecting soil and forest C stocks (see Supplementary Table S1–S2). Notably, the positive relationship between the EcM tree-dominance and all C stocks was more pronounced under unfavorable climatic conditions (significant seasonal climate fluctuation) compared

## a. Experimental design

### a1. Sampling area and ecoregions

### a2. Map of species and carbon in sampling area

## b. Hypotheses

**H1**

**H2**

**H3**

**Fig. 1 | Overview of the hypotheses. a** Experimental Design: a1, The colored map illustrates the distribution of each ecoregion, with gray points representing the distribution of sampling plots within ecoregions. a2, Map displaying Tree species richness (SR), ectomycorrhizal proportions (EcMD), arbuscular mycorrhizal proportions (AMD), tree carbon stock (TreeC), soil carbon stock (SoilC), and forest carbon stock (ForestC) at the plot scale across the sampling area. **b** Three Hypotheses: Conceptual figure illustrating the hypothetical relationships between EcM tree dominance and forest carbon stocks, as well as the regulatory effect of climate, succession (SUS), and species richness (SR) on the EcM mycorrhizal association effects. H1: Overall hypothetical relationship. H2: Effects of EcM tree dominance on forest carbon stocks weaken with increasing species richness and successional stages but strengthen with seasonal fluctuations (SF) in climate. H3: The underlying mechanisms of positive EcM mycorrhizal association effects may depend on changes in tree functional diversity (FD) along with climate, succession, and species richness. We anticipate a positive relationship between EcM tree dominance and tree/soil/forest carbon stocks, which would be more pronounced

at early than at late successional stages and at low than at high tree species diversity. At early successional stages, an increase in EcM tree dominance is expected to significantly promote forest carbon stocks due to the enhancing effect of EcM fungi on nutrient cycling, leading to increased carbon accumulation. Conversely, at late successional stages, we anticipate weaker effects of EcM mycorrhizal associations on forest carbon stocks compared to early stages, as tree interactions shift from competition to facilitation, potentially diminishing the positive effects of EcM fungal community on nutrient cycling and productivity. At low diversity levels, we expect the relationship between EcM tree dominance and carbon stocks in forests to be stronger than at high diversity, as greater niche differences at high diversity may maximize resource space occupation, thus weakening the effects of EcM tree dominance. However, under stressful conditions (significant seasonal climate fluctuations), we expect the promoting effect of EcM tree dominance on forest carbon stocks to be strengthened, while under favorable climate conditions (minimal seasonal climate fluctuation), we expect this effect to be weakened.

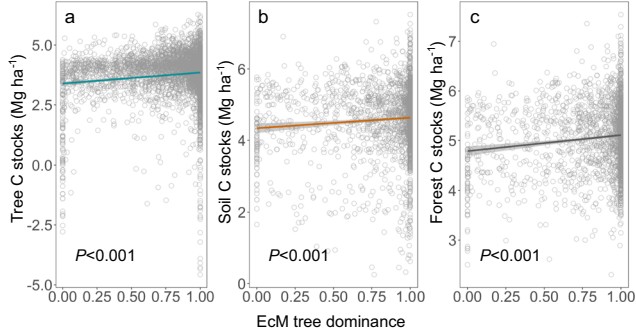

**Fig. 2 | Observed relationship between ectomycorrhizal (EcM) tree dominance and carbon stocks in forests. a** Tree carbon stocks; (**b**) soil carbon stocks; (**c**) forest carbon stocks. Tree carbon stocks, soil carbon stocks and forest carbon stocks were natural log-transformed. EcM tree dominance is quantified as the EcM tree dominance based on tree basal area. The straight lines in each figure was simple linear regression fitted across all forest plots. The solid line represents the regression fitted across all forest plots, with the solid line indicating a significant correlation ($p < 0.05$), and gray bands represent a 95% confidence interval. Each gray circle represents the data of one forest plot, with a total of 4525 plots for tree carbon stocks, 2035 plots for soil carbon stocks, and 2035 plots for forest carbon stocks.

to favorable ones (minimal seasonal climate fluctuations, Fig. 3a, d, g). We also observed a strong positive relationship between EcM tree dominance and all C stocks in the early successional stage, while the relationship weakened in the late successional stage (Fig. 3b, e, h). Furthermore, the positive relationship between EcM tree dominance and all C stocks was more pronounced at low tree species richness compared to high richness (Fig. 3c, f, i).

### Direct and indirect effects of EcM tree dominance

We employed structural equation models (SEMs) to unravel the hypothesized direct and indirect drivers and connections underlying the observed effects of EcM tree dominance on tree, soil, and forest C stocks. In line with our hypotheses, tree functional diversity emerged as the principal mediator of the indirect effects of EcM tree dominance on C stocks. However, the influence of EcM tree dominance on different C stocks varied (see Fig. 4). While EcM tree dominance directly correlated positively with tree C stocks, it did not exert an indirect influence through tree functional diversity (see Fig. 4a). Conversely, EcM tree dominance was linked to increased soil and forest C stocks, mediated by its negative impact on tree functional diversity. This negative relationship with tree functional diversity, in turn, exhibited a negative association with soil and forest C stocks (see Fig. 4b, c). Additionally, there was a notable negative relationship between EcM

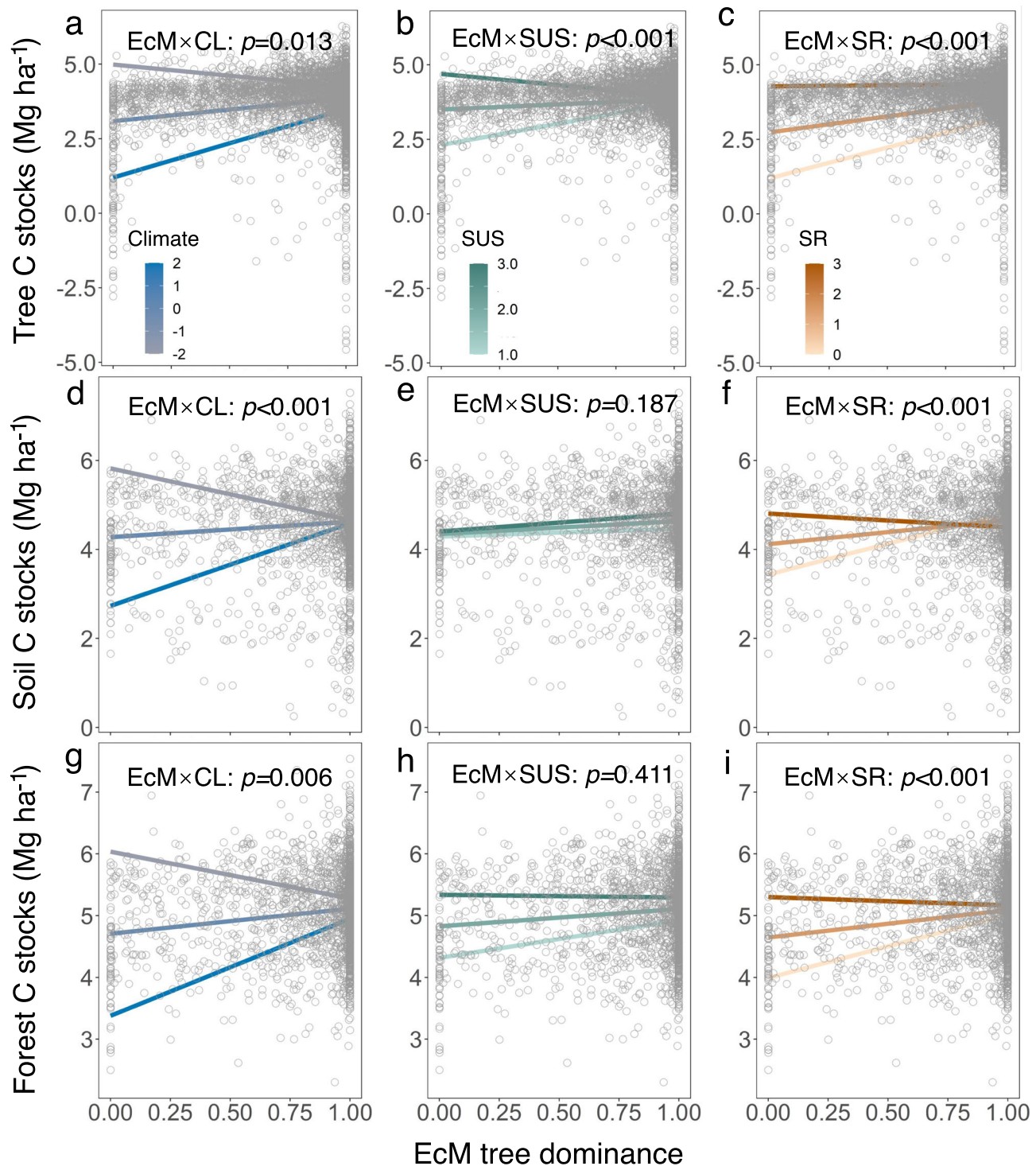

**Fig. 3 | Relationships between ectomycorrhizal (EcM) tree dominance and carbon stocks with climate (CL), different successional stages (SUS), or different species richness (SR) were examined.** Panels (**a**), (**d**), and (**g**) illustrate tree, soil and forest carbon stocks with changes in climate; panels (**b**), (**e**), and (**h**) depict tree, soil and forest carbon stocks with changes in successions; while panels (**c**), (**f**), and (**i**) show tree, soil and forest carbon stocks with changes in tree species richness. To meet the normality assumption, all carbon stock components are ln-transformed. Each dot on the plots represents a single plot. Mixed-effects models were fitted to test whether relationships between EcM tree dominance and different carbon storage components changed along the species-richness gradient, across successional stages, or with climate changes (see Supplementary Table S1 for statistical results). EcM × SR, EcM × SUS, and EcM × CL represent the interactions.

tree dominance and species richness and climate. Tree functional diversity consistently showed a strong positive relationship with climate, succession, and species richness, suggesting that climate, succession, and species richness may indirectly decrease tree and forest C stocks through their positive effect on tree functional diversity.

In our analysis of microbial communities, we found that EcM tree dominance showed a significant positive correlation with the relative abundance of soil EcM fungi and a negative correlation with the relative abundance of soil saprotrophic (SAP) fungi. Additionally, the relationship with the diversity of soil EcM and soil SAP fungi was

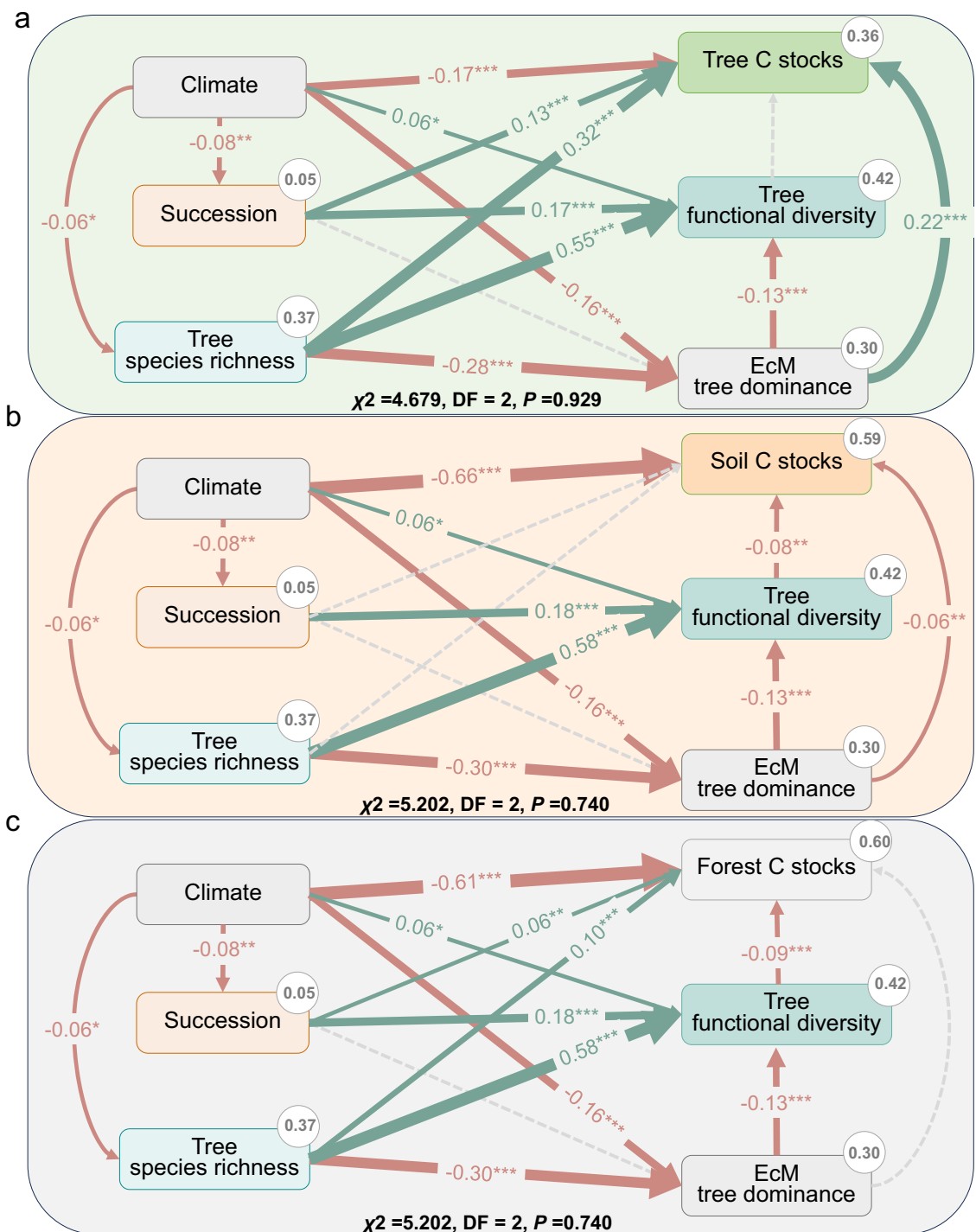

**Fig. 4 | Structural equation models (SEMs) were constructed to analyze the relationships between climate, succession, tree species richness, functional diversity, ectomycorrhizal (EcM) tree dominance, and carbon stocks in forests. a** Tree carbon stocks; (**b**), soil carbon stocks; (**c**) forest carbon stocks. Tree carbon stocks, soil carbon stocks and forest carbon stocks were natural log-transformed. Additionally, SEMs were used to assess how climatic variables, successional stages, and tree species richness modulated the effects of EcM tree dominance on carbon stocks. In the diagrams, solid blue arrows indicate positive paths ($p < 0.05$, piecewise SEM), solid red arrows represent negative paths ($p < 0.05$, $p$iecewise SEM), and solid gray arrows represent nonsignificant paths ($p > 0.05$, piecewise SEM). Path coefficients are reported as standardized effect sizes. The overall fit of the piecewise SEM was evaluated using Shipley's test of d-separation, with all models showing $p$ values > 0.05, indicating no missing paths.

inverse (see Supplementary Fig S3–S4). Overall, these results suggest that the EcM tree dominance, both directly and indirectly, may influence tree, soil, and forest C stocks mainly by enhancing tree nutrient absorption and restricting the decomposition of organic matter. This effect is particularly prominent in unfavorable climatic conditions, early successional stages, or plots with low tree species richness.

## Discussion

Our observations indicate that forests dominated by ectomycorrhizal (EcM) mycorrhizal strategies accumulate more C stocks compared to forests dominated by arbuscular mycorrhizal (AM) or mixed mycorrhizal strategies in high-latitude temperate forests. Furthermore, the beneficial effects of EcM mycorrhizal strategies on forest C stocks are

especially pronounced under unfavorable climate fluctuations, low tree species richness, or early successional stages. These findings provide insights into the patterns of forest C accumulation in high-latitude temperate forests with growth-limiting soil resources.

Consistent with our first hypothesis, forests with EcM tree dominance exhibited higher forest C stocks by increasing both tree C stocks and soil C stocks compared to forests dominated by AM or mixed mycorrhizal strategies (see Fig. 2). This finding contrasts with a recent study showing that forests with mixed mycorrhizal strategies exhibited higher productivity than those dominated by a single strategy[11]. The disparity in results may primarily stem from differences in the study areas. In middle and low latitudes, the diversity of mycorrhizal fungi may enhance the productivity of plant communities and C accumulation by facilitating plant nutrient uptake through rapid litter decomposition and complementary fungal nutrient exploitation strategies[11,37–39]. However, at relatively high latitudes, plant growth is constrained by nutrient availability, particularly soil-available N, due to slow litter decomposition[38]. In such conditions, EcM plants can access a large amount of organic N from soil organic matter (SOM) through a broad array of extracellular enzymes produced by EcM fungi[27], thereby promoting tree growth and C accumulation. Furthermore, in our study, as EcM tree dominance increased, the relative abundance of EcM fungi increased while the diversity of EcM fungi decreased. As the relative abundance of specific types of EcM fungi increases, they may form denser mycelial networks in the soil, thereby enhancing the efficiency of nutrient absorption and transfer[14,40–42], then promoting tree growth and enhancing tree C accumulation.

In addition to resource exploitation and transfer, EcM tree litters, which contain wider carbon-to-nitrogen (C: N) ratios and higher concentrations of lignin and polyphenolic compounds, have been demonstrated to decay more slowly and inhibit nutrient mineralization[14,43]. Recent studies also emphasize that EcM symbiosis dominates forests, as their fungal symbionts compete directly with free-living decomposers for N, leading to competition-induced declines in the decomposition rate of soil organic matter (SOM)[15]. Our results indicated that the relative abundance of saprotrophic (SAP) fungi was negatively correlated and significantly decreased with an increase in EcM tree dominance, indirectly suggesting that EcM tree dominance might have suppressed overall decomposition by inhibiting SAP fungi. This phenomenon may contribute to enhancing soil organic C accumulation in forest ecosystems dominated by EcM mycorrhizal strategies. In summary, the findings presented here support previous theoretical and empirical evidence suggesting that EcM symbiosis dominates forests, leading to increased soil C stocks through soil organic C stability and accumulation[15,24], particularly in N-limited high-latitude temperate forests.

Consistent with our second hypothesis, the positive impacts of EcM mycorrhizal strategies on tree, soil, and entire forest C stocks were expected to be weaker in favorable climatic conditions than in unfavorable ones and in late succession compared to early succession (see Fig. 3). Favorable climatic conditions (minimal seasonal temperature and precipitation fluctuations) may reduce the significance of EcM-associated nutrient cycling for their host trees[44], while unfavorable climatic conditions (significant seasonal temperature and precipitation fluctuations) may strengthen mycorrhizal interactions, thereby enhancing the mycorrhizal promoting effect on plant growth and forest C accumulation[31]. The primary reason for these findings is that symbiotic mycorrhizal fungi can enhance the resilience of hosts, with their influence being especially prominent under climatic conditions characterized by stress[31].

Successional patterns in mycorrhizal fungi are linked to corresponding changes in plant communities[35]. Early successional stages typically have low soil nutrient levels, prompting trees to recruit more ectomycorrhizal colonization to resist environmental pressure, such as low soil nutrients[45]. Consequently, an increase in EcM tree dominance

could enhance soil resource exploitation capacity, promoting plant growth and C accumulation in early successional stages. As the soil environment improves over time, the dependence of plants on EcM fungi to resist external environmental pressure gradually decreases[34,46,47], which may partly explain why the positive effects of EcM mycorrhizal strategies on forest C stocks are weaker in late succession than in early succession. Furthermore, in early successional stages, trees exhibit conspecific negative density dependence growth due to resource competition[48], whereas an elevated accumulation of EcM mycorrhizal fungi weakens negative density dependence[18]. This increase in EcM tree dominance can strongly promote tree coexistence, growth, and C accumulation in early successional stages. However, in late successional stages, plants demonstrate positive frequency-dependent growth[48], and the facilitative effects of tree interactions become increasingly apparent during stand development[36]. This may lead to weaker effects of EcM tree dominance in late successional stages. Overall, these results underscore the importance of ecological succession in understanding the response of forest C accumulation to mycorrhizal strategies.

Another notable observation was that the positive effects of EcM mycorrhizal strategies on tree, soil, and entire forest C stocks were weaker in species-rich stands compared to species-poor stands (see Figs. 3–4). Systematic experimental and theoretical studies have demonstrated that high tree species richness can utilize the entire resource space through various biological mechanisms, including selection effect, insurance effect, and niche complementarity effect, thereby enhancing tree growth and forest C accumulation[42,49]. In this context, the inclusion of EcM mycorrhizal strategies may offer limited advantages for enhancing resource exploitation in stands characterized by high species richness[11]. Conversely, relatively species-poor stands may achieve higher C accumulation when EcM tree species maximize resource exploitation and transfer. At high species richness levels, as communities become saturated with multiple coexisting species, the extent of niche overlap between species may increase[11,50]. Specifically, as the number of species within EcM tree groups increases, there is a higher probability that some EcM tree species will overlap in their resource utilization. This may lead to weaker effects of EcM mycorrhizal associations on tree, soil, and forest C stocks in more species-rich stands compared to less species-rich stands.

We observed a strong negative relationship between the EcM tree dominance and tree species richness. This finding aligns with evidence from an extensive grid-based inventory, demonstrating that forests dominated by EcM mycorrhizal associations tend to exhibit relatively low tree diversity[19]. These findings suggest that the effects of EcM mycorrhizal strategies on forest C stocks are not synchronized with the effects of tree species richness. Additionally, we found a negative relationship between EcM tree dominance and tree functional diversity, which was determined by tree species composition and richness. According to niche complementarity effects, the coexistence of different functional strategies should lead to fuller resource exploitation by the plant community[51]. This suggests that higher functional diversity may favor microbial activity, drive litter decomposition, and limit soil C accumulation[52]. Thus, it is plausible that EcM tree dominance enhances soil and forest C stocks indirectly through its negative effect on tree functional diversity.

Although our study has produced interesting and important results, it's crucial to recognize its inherent limitations. Firstly, our research was confined to forests in high-latitude regions, suggesting that our findings may primarily pertain to nutrient-limited environments in these areas rather than subtropical and tropical regions. Furthermore, our study relied on observational data, lacking controlled experimental evidence to substantiate our conclusions. Consequently, we intend to address this gap by conducting more controlled experiments in subsequent stages to validate our findings. It's important to acknowledge that our study's scope was limited to

specific forest types and geographical locations, potentially restricting the direct generalizability of our conclusions to other forest types or regions. Therefore, when interpreting and applying our research findings, it's imperative to carefully consider these limitations to ensure the reliability and applicability of our conclusions. Moving forward, future research endeavors should prioritize expanding the scope of investigation to explore the impacts of nutrient cycling in forests across diverse types and geographical locations, facilitating a more comprehensive understanding of the universality and complexity of this phenomenon.

In conclusion, our observation demonstrates that forests dominated by EcM strategies exhibit higher C stocks compared to those dominated by AM or mixed mycorrhizal strategies in high latitude forests, particularly under unfavorable climatic conditions, in early successional stages, and in stands with low levels of tree species richness. These findings underscore the significance of EcM mycorrhizal strategies in enhancing forest C stocks, especially in climate-sensitive and nutrient-limited high-latitude temperate forests. Our results have important implications for forest management aimed at maintaining or improving forest C stocks in high latitude forests. Given the observed negative relationship between tree species richness and EcM tree dominance, and considering that both EcM mycorrhizal strategies and species richness independently contribute to forest C stocks, we propose that planting low levels of native tree species richness with a focus on EcM mycorrhizal strategies may play a critical role in maximizing C accumulation in plantations in nutrient-limited high-latitude areas, rather than blindly increasing tree species diversity. Furthermore, our study highlights the necessity for forest management strategies to be tailored to different regional climate conditions and successional stages. By understanding the interplay between mycorrhizal strategies, species richness, succession, and climatic factors, forest managers can make informed decisions to optimize C accumulation and promote ecosystem resilience in the face of climate change.

## Methods

### Forest inventory data
For this study, we utilized data from the Forest Inventory and Analysis Program (FIAP) of Northeast China, overseen by Professor Han Shijie from 2015 to 2019 (refer to Fig. S1). The FIAP program monitors spatial patterns of C stocks in forests across Northeast China, with a sampling intensity of about one plot per 6596 hectares. Each plot measures 0.09 hectares (30 m × 30 m, totaling 900 square meters). To enhance the accuracy of tree layer assessments, each plot comprises nine smaller consecutive subplots (10 m × 10 m, totaling 100 square meters). Sampling plots were situated at least 100 m away from the nearest edge or road to minimize edge effects. All sampling plots were located within natural secondary forests and primary forests. All data collection was conducted through on-the-ground measurements.

For each plot, all tree stems with a diameter at breast height (DBH) ≥ 3 cm were measured, including coverage, DBH and height, and identified to the species level. Taxonomic names of all tree species were verified against the Flora of China (http://foc.eflora.cn/) and the Catalog of Life China (Checklist 2015, http://www.sp2000.org.cn/). Tree species richness was calculated for each plot. We employed tree species richness as a measure of local (alpha) diversity.

To describe the species composition and stability of forest ecosystems, according to Odum's theory of succession[53] and experience of scientists, we classify all sample sites of forest ecosystems into early, middle, and late stages of succession, each stage having different species composition and structure. The early stage is typically dominated by pioneer seed plants, which can grow under harsh conditions and lay the foundation for subsequent vegetation. The main types of forest communities in the early stage of succession are deciduous larch forests, birch forests, poplar forests, and mixed forests dominated by these tree species. The middle stage represents a stable phase

of succession, usually composed of mature broad-leaved trees such as Quercus, Fraxinus and Juglans trees. These tree species have relatively mild environmental requirements and can form relatively stable ecosystems. The late stage is a more complex phase, typically composed of mixed coniferous and broad-leaved trees such as Tilia, Abies and Pinus trees. We employed the compositional index proposed by Curtis and McIntosh (1951) in addition to the qualitative methods outlined above[54], to assess the community succession status. Initially, Importance Values (IV) were computed for each tree species in every plot using the formula IV = (Relative Density + Relative Canopy Cover + Relative Frequency)/3. Subsequently, we determined the top adaptation values for all tree species within the study area. Pioneer species were assigned a top adaptation value of 1, while climax species were assigned a value of 10, with other species falling within this range. Utilizing the community similarity index, we calculated similarity coefficients between other tree species and both pioneer and climax species groups to ascertain their top adaptation values[55]. The top adaptation value for each tree species was computed as (10 * similarity level to climax species group + similarity level to pioneer species group) / (similarity level to climax species group + similarity level to pioneer species group). Next, the Importance Value (IV) of each tree species in every plot was multiplied by its Climax Adaptation Value (CAV), and the sums were aggregated to derive the compositional index of the community. Sorting this compositional index in ascending order delineates the forest successional sequence community. The community successional sequence, derived from the compositional index, closely mirrors the succession sequence outlined in Odum's theory, with early succession characterized by lower compositional indices and late succession by higher indices. To simplify data interpretation, we categorized all surveyed plots into early, mid, and late successional stages according to Odum's theoretical framework.

Ecological units are defined by similarities in surficial geology, lithology, geomorphic processes, soil groups, and sub-regional climate. Using the 'National hierarchical framework of ecological units'[56], we categorized 10 ecoregions and assigned each plot to a specific ecoregion based on its geographic location.

### Tree species richness and functional diversity
We measured tree species richness by counting the number of tree species within each 900 m² plot. To assess functional diversity and identity, we considered eight key functional traits: leaf C content per leaf dry mass (C mass, mg g⁻¹), leaf nitrogen content per leaf dry mass (N mass, mg g⁻¹), leaf phosphorus content per leaf dry mass (P mass, mg g⁻¹), leaf potassium content per leaf dry mass (K mass, mg g⁻¹), average leaf area (LA, mm²), specific leaf area (SLA, mm² mg⁻¹, representing leaf area per leaf dry mass), leaf mass per area (LMA, mg mm⁻²), and leaf dry matter content (LDMC, mg g⁻¹). These traits are expected to correlate with growth and competitive abilities of species across gradients of temperature and water availability[57]. For instance, SLA is associated with plant growth rate, leaf life span, and resource uptake efficiency, while N mass and P mass are linked to plant growth and photosynthetic capacity[57,58]. We derived the trait values of C mass, N mass, P mass, K mass, LA, SLA, LMA, and LDMC using available measurements for each tree species from the TRY Plant Trait Database[59] and the China Plant Trait Database[60]. In cases where multiple data points were available for a species, we calculated the average of the available data. For species with missing data, we utilized the average trait values of all species. Functional diversity was assessed using Rao's quadratic entropy index (RaoQ), which integrates species richness and functional differences between species pairs, providing a measure of the variation in distances between species[7].

### Tree, soil and forest carbon stocks
We employed a consistent allometric equation to estimate the biomass of each tree, encompassing both above-ground and below-ground

biomass, utilizing tree height and DBH (diameter at breast height) as predictors. This approach aimed to mitigate potential artifacts arising from species-specific effects associated with the use of species-specific allometric equations[61]. To derive the allometry equation, we utilized the average coefficients from 158 sets, including those from the main dominant tree species in the region. Tree biomass estimates were then converted to tree C stocks using a conversion coefficient of 0.5, consistent with established literature[62,63]. The total tree C stock for each plot was computed by summing the tree C stocks of all individual trees. Soil samples were collected to assess soil texture, bulk density, and organic C across depth intervals of 0–10, 10–20, 20–30, 30–50, and 50–100 cm, employing a soil auger while excluding litter layers. If the soil depth of the plot was less than one meter, samples were collected from the deepest available layer. Within each depth interval, a minimum of ten samples were collected along two diagonal lines. Soil organic C and N contents were determined using a CN analyzer (multiN/C 3100, Analytik Jena AG, Germany). Site-specific soil organic C density (SOCD) for all plots was estimated by considering SOCD values across different soil depths (0–10, 10–20, 20–30, 30–50, and 50–100 cm), as described in Eq. (1):

$$SOCD = \sum (1 - V_i) \times B_i \times C_i \times T_i / 10 \quad (1)$$

In the equation provided, SOCD represents the soil organic C density (expressed in Mg C ha$^{-1}$), where: i denotes the soil layers, ranging from 0–10, 10–20, 20–30, 30–50, to 50–100 cm; Vi represents the volume percentage of gravels with a diameter exceeding 2 mm; Bi represents the bulk density (expressed in g cm$^{-3}$); Ci denotes the soil organic C content (expressed in g kg$^{-1}$); Ti represents the thickness (expressed in cm) of the ith layer.

The forest C stocks of each plot were computed by aggregating the tree C stocks and soil C stocks for each plot. Other components within each plot, such as the litter layer, shrub layer, and herb layer, were not assessed in this study. This decision was based on their relatively lower C stocks compared to tree and soil C stocks.

## Tree mycorrhizal strategy and the composition of forest mycorrhizae

Initially, the mycorrhizal strategy of each tree species was identified using a recently published database[16] that provided species-specific mycorrhizal assignments. For tree species not found in this database, the symbiotic mycorrhizal type was identified using the symbiotic fungi website (http://mycorrhizas.info/index.html) and the global online database of plant mycorrhizal associations[64]. Tree species were categorized as either ectomycorrhizal (EcM) or arbuscular mycorrhizal (AM) strategies, totaling 240 tree species. Tree species with other mycorrhizal strategies (only 1.25% of all species, i.e., ericoid mycorrhizal or non-mycorrhizal) were excluded from the analysis, as they were rare in our dataset. In cases where tree species exhibited dual EcM/AM associations, the basal area was divided equally between AM and EcM in calculating the mycorrhizal proportion[11,12]. Overall, including plant species with dual mycorrhizal types is relatively infrequent in our dataset, so their inclusion or exclusion is unlikely to significantly impact the experimental results. Total basal area for each species in each plot was computed from stem diameter measurements. Subsequently, the proportion of EcM (or AM) was computed by dividing the total basal area occupied by EcM (or AM) tree species by the total basal area of the stand in each plot.

## Climate

For each plot, temperature seasonality and precipitation seasonality were extracted from WorldClim with a resolution of 30 arc seconds, accessible at www.worldclim.org/. The two climatic factors are closely linked to mycorrhizal distribution patterns[16] and affect plant productivity[11].

## Soil fungi community

To investigate the mechanism by which EcM (or AM) proportion affects C stocks in forests, we conducted soil fungal community analysis in a subset of 110 randomly selected plots from all sampling sites. Soil samples were collected from the top 0–10 cm depth using a soil auger with a diameter of 5 cm, with ten samples collected per plot along two diagonal lines. These samples were mixed to create one composite soil sample per plot. Upon collection, soil samples were processed by sieving through a 2 mm mesh and transported to the laboratory, where DNA extraction was performed using the OMEGA Soil DNA Kit (Omega Biotek, Norcross, GA, USA) following the manufacturer's protocol. The extracted soil DNA was stored at −80 °C until further analysis. PCR amplification of the ITS1 region of the fungal 18 S rRNA gene was carried out using the primers ITS1F (5′-GGAAG TAAAAGTCGTAACAAGG-3′) and ITS2R (5′-GCTGCGTTCTTCATCGA TGC-3′). PCR amplicons were purified using Vazyme VAHTSTM DNA Clean Beads (Vazyme, Nanjing, China) and quantified with the Quant-iT PicoGreen dsDNA Assay Kit (Invitrogen, Carlsbad, CA, USA) using a Microplate reader (BioTek, FLx800). After quantification, the purified amplicons were pooled at equimolar concentrations and subjected to paired-end sequencing (2 × 250) using the MiSeq Reagent Kit V3 (600 cycles) on an Illumina MiSeq platform (Illumina Inc., San Diego, CA, USA) at Shanghai Personal Biotechnology Co., Ltd (China). The detailed methods for determining microbial communities can be found in the Supplementary Methods. The raw FASTQ files underwent quality filtering using QIIME2[65]. High-quality sequences were subsequently clustered at a 97% similarity cut-off to generate operational taxonomic units (OTUs) using Vsearch (v2.13.4). Chimeric sequences were identified and removed using UCHIME (version 4.2.40, http://drive5.com/usearch/manual/uchime algo.html). Taxonomy assignment of fungal OTUs relied on the Unite database[66], with sequences not assigned to the kingdom Fungi being discarded. Low abundance OTUs (≤10 sequences across all samples) were removed from the fungal datasets to mitigate PCR or sequencing artifacts[67]. Operational taxonomic units (OTUs) were assigned to fungal functional guilds, including EcM fungal communities and saprophytic (SAP) fungal communities[68], using the FUNGuild database (https://github.com/UMNFuN/FUNGuild). Alpha-diversity indices, specifically Shannon indexes, were calculated for all fungi, EcM fungi, and SAP fungi in each plot using MOTHUR software (version 1.30.1). These indexes provide insights into the diversity and evenness of fungal communities within each plot.

## Statistical analysis for effects of forest mycorrhizal composition on carbon stocks

We transformed the EcM tree dominance data in our inventory plots to account for the presence of only AM or only EcM trees, using a method proposed by Smithson and Verkuilen (2006)[69] and Averill et al. [70]. This transformation, denoted as y′, was calculated as y′=(y×(N-1) + 0.5)/N, where y′ represents the EcM tree dominance, bounded on the interval [0,1], and N is the sample size.

To assess the effect of EcM tree dominance on forest C storage, we developed models for tree C storage, soil C storage, and overall forest C storage. These models accounted for variations in tree species composition (tree species richness and successional stages), climate (temperature seasonality and precipitation seasonality), and topographical factors (slope and altitude). We addressed collinearity among the site descriptors by conducting principal component analysis (PCA) on the climatic variables. Axis 1, which explained 61.8% of the variance, was primarily influenced by temperature seasonality and precipitation seasonality. Introducing a novel metric, "climate", aimed at representing favorable/unfavorable climatic conditions for biological functioning, relying on temperature (TS) and precipitation seasonality (PS). The climate metric is designed to increase with concurrent rises in TS and PS values, indicating unfavorable climatic conditions if

there are significant seasonal fluctuations in temperature and precipitation. In situations characterized by significant climate fluctuations, plants may necessitate heightened adaptability to effectively manage these shifts. Conversely, smaller seasonal climate variations suggest relatively stable environmental conditions, thereby fostering more conducive circumstances for plant growth and development. Due to its effectiveness in capturing climatic conditions, we opted for the climate metric as the exclusive climatic descriptor in all models.

We initially used the Random Forest approach to identify the primary drivers of tree C storage, soil C storage, and forest C storage[71]. Then, we examined the relationship between EcM tree dominance and C storage components using simple linear regressions. Subsequently, we employed generalized linear mixed-effect models to determine the relative effects of EcM tree dominance, tree species richness, successional stages, climate, slope, and altitude on C storage. Ecoregions were included as a random effect to account for spatial heterogeneity. Additionally, we tested biologically plausible two-way interactions, such as EcM tree dominance and tree species richness, successional stages, and climate. These interactions assessed whether relationships between EcM tree dominance and C storage varied along species-richness gradients, across successional stages, or with climate changes. Model evaluation was based on the Akaike Information Criterion and R-squared values. We also examined the covariance of terms using variance inflation factors. Finally, to improve homogeneity, tree C storage, soil C storage, and forest C storage were natural log-transformed in all models.

We further evaluated the reliability of our mixed-effect models by implementing bootstrapping, a resampling technique that involves creating multiple datasets by sampling with replacement from the original dataset. Due to the uneven distribution of plot numbers across ecoregions, we grouped the data by ecoregions during the random sampling process. Specifically, we randomly selected 50% of the data from each ecoregion to generate multiple new datasets. We then constructed the same mixed-effect models using these resampled datasets and assessed their performance. The results obtained from multiple samples exhibited high consistency, indicating the robustness of our models. Therefore, we chose to present the results only once in Supplementary Table S2.

**Statistical analysis for potential causal relationships between functional diversity, climatic variables, species richness, successional stages, and forest carbon stocks**

Furthermore, we employed structural equation models (SEMs) to elucidate the direct and indirect drivers of tree, soil, and forest C stocks, guided by our a priori hypotheses rooted in existing knowledge of the mechanisms underlying the relationship between EcM tree dominance and C stocks. We investigated whether our third hypothesis was supported by examining indirect pathways that tested the effects of EcM tree dominance on C stocks mediated through functional diversity. Given that experimental manipulation of species richness can directly influence community functional diversity[72], we incorporated pathways from species richness to functional diversity and EcM tree dominance. Additionally, we employed SEMs to evaluate how successional stages and climatic modulated the effects of EcM tree dominance on C stocks. Within SEMs, we included direct pathways from climatic variables, successional stages, species richness, and EcM tree dominance to forest C stocks, testing for remaining effects not mediated by functional diversity. PiecewiseSEMs were utilized to assess the support for and relative importance of these hypothesized pathways[73], with global model fit evaluated using Fisher's C statistic ($P > 0.05$). Standardized path coefficients were calculated in the SEMs, enabling comparison of path strengths within and among models, including indirect pathways (calculated as the product of coefficients along the path). Individual pathways were fitted with linear mixed-effects models (LMMs), incorporating ecoregion as a random factor

due to its significant impact on C stocks as indicated in previous analyses. In all SEMs, tree species richness, tree C stocks, soil C stocks, and forest C stocks were log-transformed to satisfy model assumptions effectively.

**Statistical analysis for effects of forest mycorrhizal composition on fungal community**

We employed a general linear model to examine the relationship between EcM/AM tree dominance and EcM/SAP relative abundance/diversity (Supplementary Fig. S3-S4), indicating that EcM/AM tree dominance could influence C stocks in forests by modulating plant nutrient uptake and soil nutrient cycling.

Data manipulation and statistical analyses were conducted using the R platform v.4.1.1, along with the following primary packages: data.table, ggplot2, FD, sf, randomForest, gridExtra, maptools, ggpubr, ggspatial, raster, rgdal, dplyr, lme4, and piecewiseSEM.

**Reporting summary**

Further information on research design is available in the Nature Portfolio Reporting Summary linked to this article.

## Data availability

The data supporting the findings of this study can be found on Figshare via the identifier https://doi.org/10.6084/m9.figshare.25348171. The raw sequencing data were deposited in the NCBI Sequence Read Archive (SRA), SubmissionID. SUB14194707, BioProject ID. PRJNA1071870. Source data are provided with this paper.

## Code availability

The R codes supporting the findings of this study can be accessed on Figshare via the identifier https://doi.org/10.6084/m9.figshare.26114086. The code used in this work also can be accessed by contacting the corresponding authors.

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

## Acknowledgements

We are very grateful to Professor Enzai Du (State Key Laboratory of Earth Surface Processes and Resource Ecology, Faculty of Geographical Science, Beijing Normal University) for his great support and help in the preparation of the manuscript, and we are very grateful to more than 400 graduate students from eight research institutions for five consecutive years (2015–2019), which enabled us to obtain detailed data over 4000 plots. This work was supported by the National Natural Science Foundation of China (42230703, 42307160, 42377477, 41773075, 41575137, 31370494, 31170421), the Science and Technology Basic Resources Survey Special Key Project of China (2014FY110600), and the National Key Research and Development Program of China "Global Change and Response" (2016YFA0600800).

## Author contributions

Q.W., S.H. and Y.X. conceived the project. Q.W., S.H., G.Y. and Y.X. designed the study, supervised data collection, and contributed the whole manuscript preparation and design. G.Y., Q.W., Y.X., S.H., X.W., J.Z., X.L., G.H. and J.H. contributed ideas to the analysis. G.Y., Q.W., S.H. and Y.X. compiled the database. G.Y., Q.W., Y.X., J.Y., Z.G., Q.M. and J.H. analyzed the data. G.Y., Q.W., Y.X., S.H., C.F., J.Q.Z., G.L., J.Y., Z.G., W.C.L.W., W.W. Q.M., J.H.Z., Y.L., J.P.Z., X.C., X.W., L.X., Y.S., Z.Z., X.L., Y.Z., R.S., G.H., Y.F. and J.H. prepared field experiments, wrote and reviewed the manuscript, and agreed to the published version of the manuscript.

## Competing interests

The authors declare no competing interests.
