## [Peer Review File · Nature Communications]

Forest carbon stocks increase with higher dominance of ectomycorrhizal trees in high latitude forestsReviewers' comments:

Reviewer #1 (Remarks to the Author):

This manuscript addresses the relationship between the proportional abundances of ectomycorrhizal vs. arbuscular mycorrhizal-associated trees and forest carbon stocks (tree C stock + soil C stock), aiming to provide an analysis of the environmental/biotic variables that modulate this relationship at large spatial scales. The dataset collected by the authors and presented in the manuscript is very impressive, and seemingly has the potential to greatly advance our understanding of the conditions under which mycorrhizal association acts as a strong predictor of forest C storage. However, as written, the core theses and insights provided by this manuscript are challenging to assess. Some re-organization and re-writing will help to clarify the central research questions, hypotheses and approaches. I outline specific suggestions for edits below, including areas requiring more significant clarification:

Line comments:

L24-28 of Abstract: I think that the core knowledge gap this paper aims to fill could be clarified (and these sentences could be re-worded in turn). See comments throughout, especially in Introduction.

L25 ("the dominant"): Do you mean when forests are dominated by ECM-associated trees? Or when specific ECM fungi are dominant as mycorrhizal associates? After reading manuscript, I think it is the former. Please clarify.

L30: Is "forest C stock" just the sum of tree C stock + soil C stock?

L32-35: Not sure phrasing of this sentence is appropriate for an abstract. Is this a take-away from your results? Or a potential explanation that you're providing for the result in L30-31?

L36: Is this central to the results? This piece about forests "with resource limits" has not been explained yet.

L37: Unclear wording ("but tradeoffs also need to be made when")

L49-51: Seems repetitive to previous sentence

L51-53: This last sentence is also a bit unclear; the last two sentences of this paragraph might be reworded to explicitly emphasize the knowledge gap that you aim to fill here – i.e., what is already known vs. not known about the mycorrhizal control on soil/plant C storage in forests, and how does your study build on this? This last sentence just says that mycorrhizal fungi are an understudied dimension of plant functional variation, but I don't think that's accurate— can this be reworded to emphasize the knowledge gap around forest C storage?

L56: Clarify that you're referring to free-living soil microorganisms?

L60-61: Can this be reworded so it's more specific. "But their forms and functions differ greatly" is somewhat vague, and the paragraph abruptly ends with a single example illustrating this. Can you just state what the core difference is (about how they differ in their ability to acquire organic N)?

L66-68: clarify the connection here — e.g., is it due to a reduction in saprotroph biomass? Because presumably N limitation alone could drive free-living microbes to increase their enzyme production to acquire N from SOM.

L69: Replace "low" with "lower" to emphasize that the litter quality is lower (i.e., higher C:N) relative to AM tree litter.

L74 ("in N-restricted ecosystems"): Is this a qualifying statement, as in you're saying that you predict C stocks will be higher in ECM vs. AM forests, but only in N-restricted systems? This is not clear yet. After reading entire manuscript, I think that additional details need to be included in the introduction that explain the region/context in which your work is taking place (high latitude temperate forests), how this is filling a specific knowledge gap, and how this is related to your discussion of N-limited ecosystems.

L76-77: This sentence is somewhat vague, but this would be the perfect location for a clear statement about the knowledge gap(s) that your study aims to fill.

L79: Change "with" to "within"

L79-81 (starting with "few studies have quantified..."): Is this the primary focus and knowledge gap you aim to fill? After reading the entire manuscript, this feels like an important piece of it, but the intro also needs clearer statements about the geographic/biome context, what is currently known about the relationship between ECM proportional abundance and forest C storage, and all of the potential modulating variables that you aim to test (e.g., the introduction does not explain clearly enough that you are also studying tree functional diversity effects, soil N status etc.)

L83. What is meant by "mixing mycorrhizal strategies"? When both ECM and AM trees are present? What is meant by "effect strength" in this context? Please clarify wording.

L86-87: Unclear wording

L87-88: I am not sure there is support to say that these effects would be "inevitable." Please clarify the support for this statement.

L89-90: Rather than ending this paragraph by saying that there is little support for this idea, it may instead be helpful to emphasize that there is preliminary evidence that suggests this may be the case, but that it is limited in (X, Y, Z) ways. For example, based on the information you included above, it sounds like these studies may currently be restricted to single ecosystem contexts, or a couple of studies, but has not been widely tested across many forested sites (which I think is the

knowledge gap that your study aims to fill?). Or within high-latitude temperate forests?

L91: Perhaps would be clearer to say “we evaluate the relationship between ECM vs AM-associated tree abundance and forest C stocks, and whether these relationships are affected by (or modulated by) forest successional stage and tree species diversity.” It would also be helpful if you could give more context up front about how you’ll be evaluating “dominance.” Will this be treated as a continuous variable in your analyses, or as a categorical variable (AM vs. ECM dominated forests)? Should this instead be “ECM proportional abundance”?

L96: Not yet clear what is meant by “plot” in the context of your study

L98: Perhaps makes sense to just say “species richness” throughout and leave out references to “diversity” if you are not also measuring evenness?

L100-102: Not sure this sentence is needed. It seems repetitive to information included above.

L106: Unclear. Is this sentence about latitudinal effects on available soil N? Or ECM vs AM effects? Please clarify.

L107 and 100: Consider replacing references to “absorption” (or absorb) to acquire, as in ECM can acquire N from SOM through enzymatic/free radical decomposition processes.

L109: N cycling could arguably be higher in AM-dominated stands because more N is present in bioavailable forms. Please clarify the support for this hypothesis.

L106+: The other hypotheses seem to be related to AM vs ECM effects on forest C storage and how they are modulated by (2) successional stage and (3) tree species richness; thus, should hypothesis #1 be about the modulating effects of N availability on ECM vs AM relationship to forest C storage?

L111-112 (and elsewhere): Please specify what you mean by “weaker” or “stronger effects.” For example, are you hypothesizing that ECM relative abundance will be strongly positively correlated with forest C stocks?

L112-113: Consider separating out text about species richness into a separate hypothesis? The “because” statement only speaks to successional stage.

L116-118: I thought that this was the point you were making in hypothesis #1. Please clarify. Can this be moved up and combined with what you state above under H1? After seeing Fig. 1, I understand that H1 is more broadly about the relationship between ECM tree relative abundance and forest C storage, however, because the phrasing above emphasizes soil N status, this was not clear.

L118-121: Something is unclear here as well— this explanation connects most directly to the

hypotheses listed under H2. Please clarify.

L133: Since results will come first (and methods at end), describe how you are defining/calculating forest C stocks. Is this just the sum of tree C + soil C?

L133-134: Also include p-value to support this result. Here and elsewhere in the results text.

L135-136: Perhaps rephrase as “There was a significant interaction between the effect of ecoregion and ECM proportion on forest C stocks”? Please also clarify/describe how these models were set up — it sounds like there were multiple response variables considered? If so, was this a multivariate model? Or was it univariate and you looked at each of the C stock variables separately?

L142 (“significantly interacted”): See previous comment. Consider rewording this language throughout.

L144: What was the direction of the relationship? Strong positive? Apply throughout.

L144-146: Another example of why it would be helpful to provide some details on the statistical approaches/ design (since methods will come at the end of the paper). Were separate models constructed for each? Separate regression analyses run?

L159: Unclear— I don't think that you have discussed tree functional diversity yet, and whether you have a specific hypothesis about this variable (?). Should this say “tree species richness” instead? Please clarify. After reading entire manuscript, I see that you calculated a metric of tree functional diversity (based on the methods); however, this was not clear in the introduction or the background provided in the results.

L160-161: unclear wording starting at “but the influence strategies of...”

L163-164 starting with “via enhancing tree nutrient...” - Do your results present support for this? If not, perhaps leave this comment for the discussion and describe it as a possible mechanism.

L169: Please provide details on how “high species richness” vs “low species richness” plots were defined/categorized. After reading methods, it appears that plots with <5 species were considered “low richness,” however references/support is never provided for this cutoff, so it seems arbitrary. Please explain the support for this approach.

L170-172: This is a complex sentence because you are saying that within LOW richness plots, ECM relative abundance was positively correlated with species richness. How are you defining “low” richness? Since ECM proportion becomes negatively correlated with species richness within the “HIGH” richness plots, does this suggest that there is actually a hump-shaped relationship between species richness and ECM relative abundance? These sentences are made even more complex by the discussion of early vs mid successional stages, so perhaps separating these ideas

into a couple of sentences may be more straight-forward.

L177-178: Some additional clarification and description may be helpful here.

L180: How were these data collected? Has not been mentioned yet. Is this based on fungal sequencing/metabarcoding data from soils? From roots?

L182: Inversely correlated?

L183-184: Save explanation for discussion, and explain in greater detail the support for these mechanisms.

L189: I think this element of your study's focus could be emphasized more clearly in the introduction. For example, why high-latitude temperate forests? Is this an understudied ecosystem when it comes to your research questions?

L194 ("ECM tree dominance"): Here and throughout— reword to reflect what you actually measured/evaluated. I suggest something like "ECM proportional abundance was positively correlated with both tree C stocks and soil C stocks." The wording here is unclear, and I think it also does not align with how the analyses were actually conducted because the wording suggests you were looking at two categories (AM-dominated plots vs. ECM-dominated plots); however, I think all your analyses were actually continuous, correct? I made some comments related to this in the introduction— some of the phrasing could be clarified/reworded to more clearly reflect how the analyses were actually conducted.

L197: Please clarify. Higher productivity does not necessarily equate with or translate to greater forest C storage.

L215: Since your study didn't address soil C stability (only soil C stocks) it may be better to just focus your discussion on soil C stocks (quantity not stability).

L220: Within the high latitude forests that you studied, is soil moisture/ water status similar across plots? If some soils are more water-logged than others, then oxygen limitation is likely an important mechanism limiting microbial decomposition and contributing to soil C storage, which might introduce some of the variation in soil C storage observed across sites.

L285: What do these acronyms stand for?

L286: When I initially read this, I thought that your study may be relying on existing databases and some interpolation/extrapolation of values across plots, in which case this sampling intensity seems quite low. However, after reading your manuscript, my understanding was that you are just presenting data for the plots that you actually sampled on the ground. This needs to be clarified, as I am still not 100% sure of your approach, and whether this sampling intensity poses any potential limitations/issues that need to be considered when interpreting your results.

L292: Is this work that the authors of this paper conducted? Over how many years? Was all of this work conducted by on-the-ground measurements?

L295: "Extracted" makes it sound like these values were extracted from an existing database. If the author group conducted these measurements, then perhaps say that you calculated species richness, etc. for each plot.

L298-299: Since you only discuss successional stages throughout the analyses/text, perhaps describe more clearly how stand age was used in determining successional stage?

L311: This aspect of the work (on tree functional diversity) did not come through clearly in the introduction or results. Please add additional details to these sections to provide context for this work. If this is a core part of your hypotheses - that is how does tree functional diversity modulate the relationship between ECM/AM relative abundance and C storage - then I think this needs to be described more clearly in the introduction.

L323-326: Was this truly just taking an average value for C mass, N mass, P mass, SLA etc? Please provide support (references) for this approach, and for the calculation of functional diversity.

L332: Which allometric equation? Unclear.

L338: Perhaps include a brief explanation of how this method works (namely, why a coefficient of 0.5 is used, how it gives an estimate of tree C stock based on biomass etc.)

L340 (0-10cm): Were litter layers and O horizon soils removed? Does this refer just to mineral soil? If not, then methods for calculating O horizon C stocks vs. mineral soil C stocks need to be differentiated and described.

L340 (30-50, 50-100 cm depths): Was it always possible to reach these depths in every plot?

L344: What is the citation for the soil organic carbon density calculation? Throughout the paper, you refer to soil C stocks, but seemingly only calculate this metric of SOCD. Please clarify.

L367-369: This information may fit better above at L362-363 where you describe classifying trees as either AM or ECM.

L371: Maybe I am forgetting, but I don't remember climate variables being mentioned in the main text— how were these data used in your analyses?

L381: To "characterize soil fungal community composition"? More methodological details are needed to describe the fungal metabarcoding work in order for reviewer/readers to assess the quality/validity of the work.

L387: How was PCR conducted? What primers were used?

L389: Include details about bioinformatics between sequencing step and analysis of final OTU table. Also missing details about data processing steps — e.g., did you rarefy your samples to a specific sequencing depth? Did you calculate relative sequence abundances of OTUs? How many of your OTUs had functional classifications in FUNGuild? how did you treat the ones that did not in your subsequent analyses (e.g., labeling as “unknown ecology”)?

L391: Please clarify where these indices come into play in your analyses. In the main text, you emphasize “species richness”, so just alpha diversity, but not Shannon’s or Simpson’s indexes, which also take into account species evenness.

L397: I will preface this by saying that I do not have experience analyzing large datasets within the context of a study design such as this one; however, it was challenging as a reader to follow the descriptions of each of the individual models and which variables were included/excluded from each. I hope that other reviewers will be able to comment on whether this was the most robust and parsimonious approach. The Random Forest model seems useful for addressing the authors’ research questions, however additional details may be needed in the main text to clarify what was done (methods), the primary takeaways and whether they supported the results presented in the main text figures.

L421: Please provide support for this approach, otherwise the cutoff of 5 seems arbitrary. What is the ecological support for this? how common was it for the plots you studied to have <5 species?

L482-484: Unclear in this context; describe the statistical analyses that were done (e.g. ANOVA) to assess the strength/significance of these relationships

Discussion: I only made a few comments at the beginning, but decided not to make extensive comments owing to some of the clarification that is needed throughout the other sections.

Fig. 2: Throughout your figures/throughout the text, it may be clearer/more accurate to say “ECM proportional abundance” rather than dominance because dominance suggests (by definition) > 50% relative abundance. However, the scale on the x axis goes down to 0, and what you actually calculated was a proportional abundance.

Reviewer #2 (Remarks to the Author):

Review Nature Communications NCOMMS-23-58559-T

This study presents interesting results about the relationships between ectomycorrhizal fungi with carbon stocks in trees and soils in the forests of NE China, and their relationships with tree species diversity and successional stage. Such studies provide important information about ecosystem function to help guide forest management.

This study is very similar to one published recently from the US (Luo, S. et al. 2023. Higher

productivity in forests with mixed mycorrhizal strategies. *Nature Communications* 538 14(1), 1377). Although it is necessary to have studies from different parts of the world and different ecosystems to improve understanding of systems globally, the concepts and analyses are not new. The current manuscript has striking resemblances to the earlier paper.

The text of the Introduction follows closely the text and references cited in Luo et al. 2023, as well as the outline of the study and the hypotheses tested. A minor difference is that successional stage is included in the current study.

Presentation of results show many similarities with Luo et al. 2023, e.g. the conceptual figure illustrating the hypothetical relationships, the relationship between carbon stock (or productivity) with AM tree dominance, separation of low and high species richness and the threshold number of five, relative variable importance from a random forest model, structural equation models showing predictors of forest carbon stock.

Even much of the Discussion shows similarities with the text in Luo et al. 2023.

The Discussion relies too heavily on assumptions about relationships and causation without adequate evidence from the results and descriptions of the data.

The concepts of stocks and flows of C and N have been confused.

The text would benefit from a thorough editing for English expression because some sections are difficult to understand due to sentence construction and grammar.

Abstract

L25 It has long been known that fungal symbionts improve tree productivity and this is demonstrated by many of the references cited. It is unclear from this sentence what is the main point – is it that EcM are dominant over other types of fungal symbionts (arbuscular endomycorrhizae, ericoid), although commonly the type of fungi are specific to the species/ genera of host tree, rather than types being in competition.

L30 the meaning of the term ‘EcM tree dominance’ is unclear. Does it mean that the tree root system is ‘dominated’ by EcM, or that a particular EcM species is ‘dominant’, or that EcM is dominant compared with another type of fungal symbiosis?.

Introduction

L73 re-wording required ‘more C stocks’

L75, 86, 106 sentences requiring re-wording to improve clarity of English expression.

L106 High latitude is not necessarily associated with N limitation. This sentence is repeating information given in a previous paragraph.

L116-118 this sentence requires clarifying as stated as a hypothesis

Methods

L297 – 302 What is the difference between developmental stages and successional stages? Both classifications appear to be based on the stand age information, but with either 5 or 3 categories. Is there any classification of secondary and primary forest, and if so, based on what criteria? Stability is mentioned but there is no information about how this was assessed or the underlying characteristics.

Explain the nature of the succession and what changes occur in the successional stages. The forests are described as ‘natural secondary or primary’ (L290) and so would not be regenerating from bare ground. Are there changes in composition or structure of the forest? Are there related changes in the composition of EcM communities?

Refer to Figure S1 with the map showing the study region, the ecoregions and plot locations would

be useful. Comment on the spatial bias in the plot locations and the effect this has on analysis, particularly at the ecoregion level as ecoregion is one of the significant variables in the GLMs.

L336 Did the allometric equation include above and belowground biomass?

L353 The component of dead biomass should be included (dead standing trees and coarse woody debris). This component varies with stand age and can be a significant component of biomass in older stands e.g. 15-20%.

State how many plots were sampled for tree and soil carbon stocks.

L385 Explain what the second subsample was used for.

L389 define OTUs. Explain how many fungal functional guilds were used.

All sections, tables and figures in the Supplementary Material need to be referred to in the main text. Material should not be repeated.

The Supplementary Material should be separated in to Methods and Results. Results and the associated figures and tables have been included in Methods and Materials.

Section in the Methods “Soil fungal community” has much of the material repeated in the Supp. Material.

L421 What was the basis for dividing the species richness into low and high at the number of five species?

L454 Some explanation of the structural equation models is needed to show how the variables and their relationships are analysed, and hence how to interpret the results.

Figures S5 and S6, S7, S8, S9 Explain the difference between a, c, e and then b, d, f.

Soil nitrogen availability is a variable used in the results, e.g. Figure 5, but there is no mention of it being measured or the data upon which this is based. Soil N content is mentioned in the Methods as being measured by a CN analyser, but there is not further description of the data, units, depths or how this information was used. Soil N content is a very different variable to the available N in soil, which depends on the rate of mineralisation, and cannot be used interchangeably.

Results

Figure 1 is not referred to in the text and the caption seems unnecessarily long and containing material that should be in the text.

L127 Suggest that ‘most’ is inappropriate as there are high F values for a number of variables.

L129 Development stage has a higher F value than ecoregion for forest C stock.

Figure 2 l648 why is the regression reported as $p < 0.05$ when it appears to be $p < 0.001$ in Table S3?

L144 tree richness (TR) is the variable used in Table S4 not species richness. These types of inconsistencies make the paper difficult to follow.

L146 EcM proportion does not appear to be significant for late successional in any of the tables S1 – S3.

L159 ‘tree functional diversity’ is referred to in the hypotheses and the results, but it has not been defined nor explained what data are used to quantify this as a variable.

L165 The interpretation of these relationships lacks substantiation; correlations do not mean causations. A lot more experimentation would be required to demonstrate these causal relationships.

Discussion

L198 In the comparison with the results found by Luo et al. 2023, it is important to note that the

current study assessed C stocks (Mg C ha⁻¹), whereas Luo assessed productivity (Mg C ha⁻¹yr⁻¹). There are many reasons why the factors influencing a stock and a flux would be different.

L199-205 The argument about the difference due to latitude is not clear; both appear to benefit from enhancing plant nutrient uptake.

L228 Better not to use anthropomorphic terms – trees do not “allow”.

The discussion about successional stages related to soil nutrient status needs to be based on data shown in the Results. What is the difference in soil N content (or even better soil N mineralisation rate) in each of the successional stages, and the direct relationship between EcM abundance and soil N?

L276 Please clarify the relationships between the type of forests sampled in this study (natural secondary and primary forests) and the recommendations about planting forests and plantations. In providing recommendations about managing forests for climate change mitigation (the objective at the beginning of the Introduction), both the magnitude and the stability of the C stocks need to be considered and hence the role of tree species diversity.

Reviewer #3 (Remarks to the Author):

Broad comments:

The authors present data from forest inventory plots across China showing that ECM tree dominance is positively associated with soil, tree, and entire forest C stocks.

I suggest caution in the interpretation that ECM fungal associations are driving these factors—many co-varying traits of trees could be influencing these patterns, including litter chemistry, which should be included in some way in the analysis to account for the common pattern that conifers are usually ECM associated. Perhaps even more likely, these patterns may be due to climate; the ECM-dominated plots in this study appear to be clustered in the higher latitudes (Figure S2) which should have lower decomposition rates and therefore higher soil C content. Further, the implication of causality --that the mycorrhizas themselves are leading to soil C conditions-- is difficult to support, as there is the possibility that ECM associated trees are preferentially growing in places with certain soil conditions, possibly including high C stocks.

There have been several similar attempts to link ecosystem C storage with tree mycorrhizal strategies, and none are without flaw—but I fear this study is challenged by the distribution of plots along the climate gradient noted above. Without more sophisticated modeling to try to disentangle climate as a driver of soil C, it is difficult to isolate mycorrhizal type as a primary influence. The novel piece of this study is the analysis of ECM eLects on soil C along a successional gradient. I suggest re-framing this work to highlight that component of the analysis—have the main figures showing the strength of the “ECM eLect” along an x axis of increasing stand age.

I’m interested in the results of the microbial analysis as well; it’s shown here that relative abundance of saprotrophs declines with increasing ECM fungi, but what about absolute abundance? We should expect more ECM as the trees are more ECM dominated, but moving northward, we might also get an increase in the total amount of fungal biomass, including saprotrophs, which might not indicate slower decomposition caused by competition between fungal groups. This Gadgil eLect is often found most clearly in coniferous forests with very low soil and litter N, and if this is the condition of the plots in this study, litter quality and leaf habit might be

a covarying trait that is partly responsible for the patterns noted here.

Introduction: There are a lot of generalizations stated here (ECM trees have low litter quality, ECM fungi produce enzymes that liberate SOM N, ECM forests store more C...) that are much more nuanced than they are portrayed in this section. The literature cited here is often the first study on a certain topic, but for all these claims, many more studies have

followed showing that the truth is much more complex. I'd suggest moderating the language to indicate that these things are true in some forests and some contexts, but not always.

Line by line comments

Line 61: Some ECM taxa can produce these enzymes, but not all can. See Pellitier and Zak, 2018.

Line 68: There has been a lot of discussion about this idea, showing this is not always the case—please look at studies that followed on the ones cited here to get a more balanced picture of this process.

Line 69: Not all ECM trees produce low quality litter, this is a generalization that is often true but in some forests (birch is a good example) ECM trees have quite labile litter. Differences in litter quality between AM and ECM tree species also decrease as you look closer to the tropics (see Keller and Phillips, 2019).

Line 85-88: This sentence is unclear, but I think it is important to express this idea with more details because it's not clear as yet why you expect species richness of the trees to mediate mycorrhizal association eLects.

Line 106-109: Meaning is not clear here.

Line 113-114: Why do you expect this shift from competition to facilitation? I'd like to hear more rationale for this earlier in the introduction. It is also not clear why this shift would then lead to lower impacts of ECM dominance.

Lines 127 & 129: Given that the results come before Methods in this publication, it would be nice to get some brief description (very short) of what you mean by "development stage" and "ecoregion", just so we know what you're presenting in this initial section. Even just "forest development stage (i.e. early successional, mid-successional, or late successional)" or something like that.

Line 158: typo: "the observed"

Line 160: What is meant by influence strategies?

Line 182: I would describe the sign of the relationship (positive/negative) instead of using "inversely"

Line 183: I would say "may influence tree and..." because these are just correlations, and we can't say anything about causation.

Line 198-199: I appreciate the note that this result is inconsistent with the dataset from the U.S. FIA plots and it seems plausible that latitude and climate/soil conditions could be partly to blame, so I suggest noting the latitude ranges of the plots in each study. Interestingly, figure S2 shows that this latitude range pattern also applies to the plots used in this study alone—there are a lot more ECM dominated plots in general, and they are clustered more in the northern part of the region. How do you know climate isn't the driving force behind the higher soil C stocks there? In truth, I think we know it's not just one driver, and these are all connected; cooler forests tend to have slower decomposition, resulting in low N availability; this favors ECM-associated plants, some of which can mine nitrogen from particulate organic matter, and this leads to trends in the distribution of AM

and ECM associated trees globally. But this means we should be very cautious about picking out the mycorrhizal fungi as drivers of soil conditions, when their presence in certain forest types is likely ultimately caused by the overlying climate conditions. They may reinforce these soil conditions, but their dominance is likely a symptom of the initial low N status of northern forests. Figures: The colors and formatting look very nice; the figures are clear and informative. Figure S2 is important and I believe should be in the main text.

Reviewer #1 (Responses in BLUE):

This manuscript addresses the relationship between the proportional abundances of ectomycorrhizal vs. arbuscular mycorrhizal-associated trees and forest carbon stocks (tree C stock + soil C stock), aiming to provide an analysis of the environmental/biotic variables that modulate this relationship at large spatial scales. The dataset collected by the authors and presented in the manuscript is very impressive, and seemingly has the potential to greatly advance our understanding of the conditions under which mycorrhizal association acts as a strong predictor of forest C storage. However, as written, the core theses and insights provided by this manuscript are challenging to assess. Some re-organization and re-writing will help to clarify the central research questions, hypotheses and approaches. I outline specific suggestions for edits below, including areas requiring more significant clarification:

Thank you for your valuable feedback on our manuscript. We appreciate your acknowledgment of the impressive dataset we have presented and the potential impact of our study on advancing understanding of the relationship between mycorrhizal associations and forest carbon stocks. We will carefully consider your suggestions for reorganizing and rewriting the manuscript to enhance clarity, particularly in articulating the central research questions, hypotheses, and approaches. Your guidance will undoubtedly help us improve the overall structure and readability of the manuscript, ensuring it meets the standards of Nature Communications.

Line comments (Responses in BLUE):

L24-28 of Abstract: I think that the core knowledge gap this paper aims to fill could be clarified (and these sentences could be re-worded in turn). See comments throughout, especially in Introduction.

Thank you for your insights regarding the abstract. We acknowledge the need to clarify the core knowledge gap addressed by our paper and will work on refining these points, as well as re-wording the sentences accordingly. We will pay close attention to your comments throughout the Introduction section to ensure a clearer

articulation of our research aims and the gap we aim to fill. Moreover, in the revised manuscript, this abstract was rewritten.

L25 ("the dominant"): Do you mean when forests are dominated by ECM-associated trees? Or when specific ECM fungi are dominant as mycorrhizal associates? After reading manuscript, I think it is the former. Please clarify.

Thank you for your question. In our manuscript, "the dominant" refers to forests that are dominated by ectomycorrhizal (ECM)-associated trees. We explained this in the revised manuscript (see Line 24-25).

L30: Is "forest C stock" just the sum of tree C stock + soil C stock?

Yes

L32-35: Not sure phrasing of this sentence is appropriate for an abstract. Is this a take-away from your results? Or a potential explanation that you're providing for the result in L30-31?

In the revised manuscript, this sentence has been rephrased (see Line 32).

L36: Is this central to the results? This piece about forests "with resource limits" has not been explained yet.

In the revised manuscript, "with resource limits" has been deleted (see Line 33).

L37: Unclear wording ("but tradeoffs also need to be made when")

In the revised manuscript, this sentence has been rephrased (see Line 33-37).

L49-51: Seems repetitive to previous sentence

In the revised manuscript, this sentence has been deleted (see Line 46).

L51-53: This last sentence is also a bit unclear; the last two sentences of this paragraph might be reworded to explicitly emphasize the knowledge gap that you aim to fill here – i.e., what is already known vs. not known about the mycorrhizal control on soil/plant C storage in forests, and how does your study build on this? This last sentence just says that mycorrhizal fungi are an understudied dimension of plant functional variation, but I don't think that's accurate— can this be reworded to emphasize the knowledge gap around forest C storage?

This sentence has been rephrased in the revised manuscript (see Line 47).

L56: Clarify that you're referring to free-living soil microorganisms?

Yes, the word has been revised (see Line 50).

L60-61: Can this be reworded so it's more specific. "But their forms and functions differ greatly" is somewhat vague, and the paragraph abruptly ends with a single example illustrating this. Can you just state what the core difference is (about how they differ in their ability to acquire organic N)?

The sentence has been revised (see Line 55).

L66-68: clarify the connection here — e.g., is it due to a reduction in saprotroph biomass? Because presumably N limitation alone could drive free-living microbes to increase their enzyme production to acquire N from SOM.

The sentence has been revised (see Line 61).

L69: Replace "low" with "lower" to emphasize that the litter quality is lower (i.e., higher C:N) relative to AM tree litter.

Done (see Line 65).

L74 ("in N-restricted ecosystems"): Is this a qualifying statement, as in you're saying that you predict C stocks will be higher in ECM vs. AM forests, but only in N-restricted systems? This is not clear yet. After reading entire manuscript, I think that

additional details need to be included in the introduction that explain the region/context in which your work is taking place (high latitude temperate forests), how this is filling a specific knowledge gap, and how this is related to your discussion of N-limited ecosystems.

This passage has been rewritten in the revised manuscript.

L76-77: This sentence is somewhat vague, but this would be the perfect location for a clear statement about the knowledge gap(s) that your study aims to fill.

The sentence has been revised (see Line 72).

L79: Change “with” to “within”

Done (see Line 75).

L79-81 (starting with “few studies have quantified...”): Is this the primary focus and knowledge gap you aim to fill? After reading the entire manuscript, this feels like an important piece of it, but the intro also needs clearer statements about the geographic/biome context, what is currently known about the relationship between ECM proportional abundance and forest C storage, and all of the potential modulating variables that you aim to test (e.g., the introduction does not explain clearly enough that you are also studying tree functional diversity effects, soil N status etc.)

Thank you very much for your feedback. We have made extensive revisions to the content of the introduction, aiming to clearly articulate the purpose of our research in the revised manuscript.

L83. What is meant by “mixing mycorrhizal strategies”? When both ECM and AM trees are present? What is meant by “effect strength” in this context? Please clarify wording.

The sentence had been revised (see Line 80).

L86-87: Unclear wording

The sentence has been revised (see Line 82-84).

L87-88: I am not sure there is support to say that these effects would be “inevitable.” Please clarify the support for this statement.

The sentence has been revised (see Line 84-85).

L89-90: Rather than ending this paragraph by saying that there is little support for this idea, it may instead be helpful to emphasize that there is preliminary evidence that suggests this may be the case, but that it is limited in (X, Y, Z) ways. For example, based on the information you included above, it sounds like these studies may currently be restricted to single ecosystem contexts, or a couple of studies, but has not been widely tested across many forested sites (which I think is the knowledge gap that your study aims to fill?). Or within high-latitude temperate forests?

Yes, this has not been widely tested across many forested sites. The sentence has been revised (see Line 84-87).

L91: Perhaps would be clearer to say “we evaluate the relationship between ECM vs AM-associated tree abundance and forest C stocks, and whether these relationships are affected by (or modulated by) forest successional stage and tree species diversity.” It would also be helpful if you could give more context up front about how you’ll be evaluating “dominance.” Will this be treated as a continuous variable in your analyses, or as a categorical variable (AM vs. ECM dominated forests)? Should this instead be “ECM proportional abundance”?

Yes, the sentence has been revised (see Line 88-90).

L96: Not yet clear what is meant by “plot” in the context of your study

The information has been added (see Line 93).

L98: Perhaps makes sense to just say “species richness” throughout and leave out references to “diversity” if you are not also measuring evenness?

The sentence has been revised (see Line 95).

L100-102: Not sure this sentence is needed. It seems repetitive to information included above.

This sentence provides additional explanation of the concept of succession, which we believe should be retained to assist readers in understanding.

L106: Unclear. Is this sentence about latitudinal effects on available soil N? Or ECM vs AM effects? Please clarify.

The sentence has been revised (see Line 105).

L107 and 100: Consider replacing references to “absorption” (or absorb) to acquire, as in ECM can acquire N from SOM through enzymatic/free radical decomposition processes.

Done (see Line 105).

L109: N cycling could arguably be higher in AM-dominated stands because more N is present in bioavailable forms. Please clarify the support for this hypothesis.

Done (see Line 104-106).

L106+: The other hypotheses seem to be related to AM vs ECM effects on forest C storage and how they are modulated by (2) successional stage and (3) tree species richness; thus, should hypothesis #1 be about the modulating effects of N availability on ECM vs AM relationship to forest C storage?

Thank you for your feedback. All hypotheses have been rewritten in the revised manuscript (see Line 106-118).

L111-112 (and elsewhere): Please specify what you mean by “weaker” or “stronger effects.” For example, are you hypothesizing that ECM relative abundance will be strongly positively correlated with forest C stocks?

Thank you for your feedback. All hypotheses have been rewritten in the revised manuscript (see Line 106-118).

L112-113: Consider separating out text about species richness into a separate hypothesis? The “because” statement only speaks to successional stage.

Done (see Line 109-111).

L116-118: I thought that this was the point you were making in hypothesis #1. Please clarify. Can this be moved up and combined with what you state above under H1? After seeing Fig. 1, I understand that H1 is more broadly about the relationship between ECM tree relative abundance and forest C storage, however, because the phrasing above emphasizes soil N status, this was not clear.

The sentence has been revised (see Line 116-118).

L118-121: Something is unclear here as well— this explanation connects most directly to the hypotheses listed under H2. Please clarify.

The H2 has been revised (see Line 109-111).

L133: Since results will come first (and methods at end), describe how you are defining/calculating forest C stocks. Is this just the sum of tree C + soil C?

The sentence has been revised (see Line 127-128).

L133-134: Also include p-value to support this result. Here and elsewhere in the results text.

Done (see Line 128).

L135-136: Perhaps rephrase as “There was a significant interaction between the effect of ecoregion and ECM proportion on forest C stocks”? Please also clarify/describe how these models were set up — it sounds like there were multiple response variables

considered? If so, was this a multivariate model? Or was it univariate and you looked at each of the C stock variables separately?

We have altered the statistical methodology and made significant modifications to the description of the results. Please review the revised results.

L142 (“significantly interacted”): See previous comment. Consider rewording this language throughout.

The sentence has been revised (see Line 138).

L144: What was the direction of the relationship? Strong positive? Apply throughout.

Done (see Line 141).

L144-146: Another example of why it would be helpful to provide some details on the statistical approaches/ design (since methods will come at the end of the paper). Were separate models constructed for each? Separate regression analyses run?

We have altered the statistical methodology and made significant modifications to the description of the results. Please review the revised results.

L159: Unclear— I don't think that you have discussed tree functional diversity yet, and whether you have a specific hypothesis about this variable (?). Should this say “tree species richness” instead? Please clarify. After reading entire manuscript, I see that you calculated a metric of tree functional diversity (based on the methods); however, this was not clear in the introduction or the background provided in the results.

Thank you for your feedback. We have made extensive revisions to this section and aim for the modified content to clearly convey our intended meaning.

L160-161: unclear wording starting at “but the influence strategies of...”

The sentence has been revised (see Line 155).

L163-164 starting with “via enhancing tree nutrient...” - Do your results present support for this? If not, perhaps leave this comment for the discussion and describe it as a possible mechanism.

The sentence has been deleted (see Line 156).

L169: Please provide details on how “high species richness” vs “low species richness” plots were defined/categorized. After reading methods, it appears that plots with <5 species were considered “low richness,” however references/support is never provided for this cutoff, so it seems arbitrary. Please explain the support for this approach.

In the initial data analysis, we referenced the method proposed by Luo et al (2023). We discussed the impact of species richness on carbon storage using categorical variables. In the revised manuscript, we have analyzed species richness as a continuous variable. For detailed information, please refer to the revised manuscript.

L170-172: This is a complex sentence because you are saying that within LOW richness plots, ECM relative abundance was positively correlated with species richness. How are you defining “low” richness? Since ECM proportion becomes negatively correlated with species richness within the “HIGH” richness plots, does this suggest that there is actually a hump-shaped relationship between species richness and ECM relative abundance? These sentences are made even more complex by the discussion of early vs mid successional stages, so perhaps separating these ideas into a couple of sentences may be more straight-forward.

Thank you for your feedback. We also agree that treating species richness as a categorical variable may not be entirely appropriate. Therefore, we have revised our analytical approach accordingly. Please review the revised manuscript for the updated analysis.

L177-178: Some additional clarification and description may be helpful here.

Done (see Line 164).

L180: How were these data collected? Has not been mentioned yet. Is this based on fungal sequencing/metabarcoding data from soils? From roots?

Due to limitations in the word count, we have provided details of the microbial soil sampling and sequencing methods in the appendix.

L182: Inversely correlated?

Yes.

L183-184: Save explanation for discussion, and explain in greater detail the support for these mechanisms.

Done.

L189: I think this element of your study's focus could be emphasized more clearly in the introduction. For example, why high-latitude temperate forests? Is this an understudied ecosystem when it comes to your research questions?

Thank you very much for your feedback. We completely agree with your suggestion, and we have included the relevant information in the revised manuscript.

L194 ("ECM tree dominance"): Here and throughout— reword to reflect what you actually measured/evaluated. I suggest something like "ECM proportional abundance was positively correlated with both tree C stocks and soil C stocks." The wording here is unclear, and I think it also does not align with how the analyses were actually conducted because the wording suggests you were looking at two categories (AM-dominated plots vs. ECM-dominated plots); however, I think all your analyses were actually continuous, correct? I made some comments related to this in the introduction— some of the phrasing could be clarified/reworded to more clearly reflect how the analyses were actually conducted.

Thank you for your feedback. We have standardized the wording in the revised manuscript to enhance readability. Please review the revised draft for further inspection.

L197: Please clarify. Higher productivity does not necessarily equate with or translate to greater forest C storage.

Yes, the sentence has been revised (see Line 182).

L215: Since your study didn't address soil C stability (only soil C stocks) it may be better to just focus your discussion on soil C stocks (quantity not stability).

Yes, I completely agree with your point of view, and this sentence has been modified (see Line 201-203).

L220: Within the high latitude forests that you studied, is soil moisture/ water status similar across plots? If some soils are more water-logged than others, then oxygen limitation is likely an important mechanism limiting microbial decomposition and contributing to soil C storage, which might introduce some of the variation in soil C storage observed across sites.

I agree with your comment. The surveyed area rarely experiences waterlogged soil; hence, we did not consider the influence of soil moisture.

L285: What do these acronyms stand for?

The sentence has been revised (see Line 287).

L286: When I initially read this, I thought that your study may be relying on existing databases and some interpolation/extrapolation of values across plots, in which case this sampling intensity seems quite low. However, after reading your manuscript, my understanding was that you are just presenting data for the plots that you actually sampled on the ground. This needs to be clarified, as I am still not 100% sure of your approach, and whether this sampling intensity poses any potential limitations/issues that need to be considered when interpreting your results.

We obtained the data through extensive surveys, and in the revised manuscript, we strive to elucidate the data collection methods clearly. If you have any suggestions for improvement, we would greatly appreciate your input.

L292: Is this work that the authors of this paper conducted? Over how many years? Was all of this work conducted by on-the-ground measurements?

This project spanned five years, during which over 400 teachers and students participated in the survey work. Our authors include all the key members of the project, and in the acknowledgments, we express our gratitude to all the teachers and students involved in the project.

L295: “Extracted” makes it sound like these values were extracted from an existing database. If the author group conducted these measurements, then perhaps say that you calculated species richness, etc. for each plot.

Done (see Line 298).

L298-299: Since you only discuss successional stages throughout the analyses/text, perhaps describe more clearly how stand age was used in determining successional stage?

In the new analysis, we did not consider the influence of age but instead focused on the changes in species composition to characterize successional dynamics (see Line 299-303).

L311: This aspect of the work (on tree functional diversity) did not come through clearly in the introduction or results. Please add additional details to these sections to provide context for this work. If this is a core part of your hypotheses - that is how does tree functional diversity modulate the relationship between ECM/AM relative abundance and C storage - then I think this needs to be described more clearly in the introduction.

We agree with your viewpoint. We have added relevant information to the introduction.

L323-326: Was this truly just taking an average value for C mass, N mass, P mass, SLA etc? Please provide support (references) for this approach, and for the calculation of functional diversity.

The reference for this method has been added (see Line 327).

L338: Perhaps include a brief explanation of how this method works (namely, why a coefficient of 0.5 is used, how it gives an estimate of tree C stock based on biomass etc.)

The reference for this method has been added (see Line 337).

L340 (0-10cm): Were litter layers and O horizon soils removed? Does this refer just to mineral soil? If not, then methods for calculating O horizon C stocks vs. mineral soil C stocks need to be differentiated and described.

Yes, litter layers were removed. Soil samples were collected to assess soil texture, bulk density, and organic C across depth intervals of 0-10, 10-20, 20-30, 30-50, and 50-100 cm, employing a soil auger while excluding litter layers. We measured the bulk density of each soil layer separately and calculated their carbon content individually. The sum of these values equals the total soil carbon content.

L340 (30-50, 50-100 cm depths): Was it always possible to reach these depths in every plot?

If the soil depth of the plot was less than one meter, samples were collected from the deepest available layer.

L344: What is the citation for the soil organic carbon density calculation? Throughout the paper, you refer to soil C stocks, but seemingly only calculate this metric of SOCD. Please clarify.

In many studies, carbon density is commonly used to characterize carbon pools.

L367-369: This information may fit better above at L362-363 where you describe classifying trees as either AM or ECM.

Thank you for your suggestion; this sentence has been modified (see Line 371-372).

L371: Maybe I am forgetting, but I don't remember climate variables being mentioned in the main text— how were these data used in your analyses?

In the revised manuscript, we have taken into account the impact of climate on carbon stocks (see Line 376-379).

L381: To “characterize soil fungal community composition”? More methodological details are needed to describe the fungal metabarcoding work in order for reviewer/readers to assess the quality/validity of the work.

Due to the word limit of the manuscript, detailed methodological steps are presented in the supporting materials.

L387: How was PCR conducted? What primers were used?

Due to the word limit of the manuscript, detailed methodological steps are presented in the supporting materials.

L389: Include details about bioinformatics between sequencing step and analysis of final OTU table. Also missing details about data processing steps — e.g., did you rarefy your samples to a specific sequencing depth? Did you calculate relative sequence abundances of OTUs? How many of your OTUs had functional classifications in FUNGuild? how did you treat the ones that did not in your subsequent analyses (e.g., labeling as “unknown ecology”)?

Detailed methodological steps for microbial community determination are presented in the supporting materials.

L391: Please clarify where these indices come into play in your analyses. In the main text, you emphasize “species richness”, so just alpha diversity, but not Shannon’s or Simpson’s indexes, which also take into account species evenness.

All results of microbial community data analysis are presented in the supporting materials and are not shown in the main text.

L397: I will preface this by saying that I do not have experience analyzing large datasets within the context of a study design such as this one; however, it was challenging as a reader to follow the descriptions of each of the individual models and which variables were included/excluded from each. I hope that other reviewers will be able to comment on whether this was the most robust and parsimonious approach. The Random Forest model seems useful for addressing the authors’ research questions, however additional details may be needed in the main text to clarify what was done (methods), the primary takeaways and whether they supported the results presented in the main text figures.

I completely agree with your opinion. During the revision process, we simplified the data classification method in the manuscript to enhance the readability of data processing and analysis.

L421: Please provide support for this approach, otherwise the cutoff of 5 seems arbitrary. What is the ecological support for this? how common was it for the plots you studied to have <5 species?

We refer to Luo et al. (2023) method for analyzing species richness as a taxonomic variable. Based on the reviewer's feedback, it is widely agreed that the classification lacks a basis. Therefore, in the revised manuscript, we will analyze species richness as a continuous variable.

L482-484: Unclear in this context; describe the statistical analyses that were done (e.g. ANOVA) to assess the strength/significance of these relationships

All data processing methods have been revised. Please review the revised manuscript for verification.

Discussion: I only made a few comments at the beginning, but decided not to make extensive comments owing to some of the clarification that is needed throughout the other sections.

Thank you very much for your feedback. We have made extensive revisions to the manuscript based on your comments, resulting in a significant improvement in its quality.

Fig. 2: Throughout your figures/throughout the text, it may be clearer/more accurate to say “ECM proportional abundance” rather than dominance because dominance suggests (by definition) > 50% relative abundance. However, the scale on the x axis goes down to 0, and what you actually calculated was a proportional abundance.

I agree with your point. We have defined dominance in the caption, aiming for clearer expression of our viewpoint in the revised manuscript.

Reviewer #2 (Responses in BLUE):

Review Nature Communications NCOMMS-23-58559-T

This study presents interesting results about the relationships between ectomycorrhizal fungi with carbon stocks in trees and soils in the forests of NE China, and their relationships with tree species diversity and successional stage. Such studies provide important information about ecosystem function to help guide forest management.

Thank you very much for the reviewer's recognition of our manuscript.

This study is very similar to one published recently from the US (Luo, S. et al. 2023. Higher productivity in forests with mixed mycorrhizal strategies. Nature Communications 538 14(1), 1377). Although it is necessary to have studies from

different parts of the world and different ecosystems to improve understanding of systems globally, the concepts and analyses are not new. The current manuscript has striking resemblances to the earlier paper. The text of the Introduction follows closely the text and references cited in Luo et al. 2023, as well as the outline of the study and the hypotheses tested. A minor difference is that successional stage is included in the current study. Presentation of results show many similarities with Luo et al. 2023, e.g. the conceptual figure illustrating the hypothetical relationships, the relationship between carbon stock (or productivity) with AM tree dominance, separation of low and high species richness and the threshold number of five, relative variable importance from a random forest model, structural equation models showing predictors of forest carbon stock. Even much of the Discussion shows similarities with the text in Luo et al. 2023. The Discussion relies too heavily on assumptions about relationships and causation without adequate evidence from the results and descriptions of the data. The concepts of stocks and flows of C and N have been confused.

Thank you for your feedback. As you mentioned, we referenced Luo et al. (2023) data processing method during our analysis. Initially, we aimed to compare the differences in mycorrhizal effects between high-latitude regions and the mid-low latitude regions studied by Luo et al. (2023). Therefore, we adopted consistent methods for comparability between the two studies. However, during the peer-review process, concerns were raised regarding the treatment of richness as a categorical variable. Consequently, we revised all data processing methods. Furthermore, the editor suggested focusing on the regulation of mycorrhizal effects by climate and succession. As a result, we made extensive modifications to the manuscript. We hope these revisions enhance the scientific value of the manuscript.

The text would benefit from a thorough editing for English expression because some sections are difficult to understand due to sentence construction and grammar.

Thank you for your feedback. We have had native English speakers review and edit the language of our manuscript. We hope that the revised manuscript meets the requirements of the journal.

Abstract

L25 It has long been known that fungal symbionts improve tree productivity and this is demonstrated by many of the references cited. It is unclear from this sentence what is the main point – is it that EcM are dominant over other types of fungal symbionts (arbuscular endomycorrhizae, ericoid), although commonly the type of fungi are specific to the species/ genera of host tree, rather than types being in competition.

The sentence had been revised (see Line 24-25).

L30 the meaning of the term ‘EcM tree dominance’ is unclear. Does it mean that the tree root system is ‘dominated’ by EcM, or that a particular EcM species is ‘dominant’, or that EcM is dominant compared with another type of fungal symbiosis?.

The mycorrhizal dominance has been defined in abstract (see Line 28).

Introduction

L73 re-wording required ‘more C stocks’

Done (see Line 68).

L75, 86, 106 sentences requiring re-wording to improve clarity of English expression.

Done (see Line 68, 82, and 104).

L106 High latitude is not necessarily associated with N limitation. This sentence is repeating information given in a previous paragraph.

The sentence has been revised (see Line 104).

L116-118 this sentence requires clarifying as stated as a hypothesis

Methods

Done (see Line 116-118).

L297-302 What is the difference between developmental stages and successional stages? Both classifications appear to be based on the stand age information, but with either 5 or 3 categories. Is there any classification of secondary and primary forest, and if so, based on what criteria? Stability is mentioned but there is no information about how this was assessed or the underlying characteristics. Explain the nature of the succession and what changes occur in the successional stages. The forests are described as 'natural secondary or primary' (1290) and so would not be regenerating from bare ground. Are there changes in composition or structure of the forest? Are there related changes in the composition of EcM communities?

In the new methodology, we did not consider the developmental stages of the community (mainly determined by age) but focused solely on the successional stages. Different successional stages primarily influence plant species composition, which in turn affects the proportion of mycorrhizal species.

Refer to Figure S1 with the map showing the study region, the ecoregions and plot locations would be useful. Comment on the spatial bias in the plot locations and the effect this has on analysis, particularly at the ecoregion level as ecoregion is one of the significant variables in the GLMs.

We agree with your suggestion. In the revised manuscript, we have included the distribution of sampling points in Figure 1. Additionally, we have altered the data analysis method by employing a mixed-model approach, where ecological functional zones are treated as random factors, allowing us to mitigate the effects of ecological functional zone heterogeneity.

L336 Did the allometric equation include above and belowground biomass?

The aboveground and belowground allometric equations are different.

L353 The component of dead biomass should be included (dead standing trees and coarse woody debris). This component varies with stand age and can be a significant component of biomass in older stands e.g. 15-20%. State how many plots were sampled for tree and soil carbon stocks.

We agree with your point. The component of dead biomass is indeed an important component of forest carbon storage. However, we did not focus on it during our survey, so it cannot be included in our analysis. We will consider the issue you mentioned in our future research.

L385 Explain what the second subsample was used for.

This sentence was inaccurately expressed and has been revised (see Line 388).

L389 define OTUs. Explain how many fungal functional guilds were used. All sections, tables and figures in the Supplementary Material need to be referred to in the main text. Material should not be repeated. The Supplementary Material should be separated in to Methods and Results. Results and the associated figures and tables have been included in Methods and Materials. Section in the Methods “Soil fungal community” has much of the material repeated in the Supp. Material.

Yes, due to constraints in length and word count, we have placed a portion of the methods for fungal community determination and analysis in the supporting materials. If necessary, we can consider moving some of the methods from the supporting materials to the main text in the future.

L421 What was the basis for dividing the species richness into low and high at the number of five species?

In the initial analysis, we referenced Luo et al. (2023) method, treating species richness as a categorical variable. However, we now recognize the lack of basis for this classification approach. Therefore, in the new methodology, we have considered species richness as a continuous variable. Please review the revised manuscript for verification.

L454 Some explanation of the structural equation models is needed to show how the variables and their relationships are analysed, and hence how to interpret the results. Figures S5 and S6, S7, S8, S9 Explain the difference between a, c, e and then b, d, f. Soil nitrogen availability is a variable used in the results, e.g. Figure 5, but there is no mention of it being measured or the data upon which this is based. Soil N content is mentioned in the Methods as being measured by a CN analyser, but there is not further description of the data, units, depths or how this information was used. Soil N content is a very different variable to the available N in soil, which depends on the rate of mineralisation, and cannot be used interchangeably.
We completely agree with your point. Causal relationships are essential in structural equation model construction. Therefore, in the revised manuscript, we have removed soil nitrogen as a factor from the structural equation model, making it simpler and easier to understand.

Results

Figure 1 is not referred to in the text and the caption seems unnecessarily long and containing material that should be in the text.

The content has been revised accordingly.

L127 Suggest that 'most' is inappropriate as there are high F values for a number of variables.

The content has been revised (see Line 124).

L129 Development stage has a higher F value than ecoregion for forest C stock.

Development stage has been deleted (see Line 126).

Figure 2 1648 why is the regression reported as $p < 0.05$ when it appears to be $p < 0.001$ in Table S3?

The content has been revised.

L144 tree richness (TR) is the variable used in Table S4 not species richness. These types of inconsistencies make the paper difficult to follow.

The content has been revised. Please review the revised manuscript for verification.

L146 EcM proportion does not appear to be significant for late successional in any of the tables S1 – S3.

The content has been modified. Please review the revised manuscript.

L159 ‘tree functional diversity’ is referred to in the hypotheses and the results, but it has not been defined nor explained what data are used to quantify this as a variable.

We introduced the method of calculating functional diversity and its sources in the methods section (see Line 325).

L165 The interpretation of these relationships lacks substantiation; correlations do not mean causations. A lot more experimentation would be required to demonstrate these causal relationships.

We agree with your point. Indeed, our research lacks mechanistic investigation. However, it cannot be denied that we have also presented significant findings.

Discussion

L198 In the comparison with the results found by Luo et al. 2023, it is important to note that the current study assessed C stocks (Mg C ha⁻¹), whereas Luo assessed productivity (Mg C ha⁻¹yr⁻¹). There are many reasons why the factors influencing a stock and a flux would be different.

In most cases, we tend to believe that higher forest productivity may lead to larger carbon stocks in forests. However, there are certainly exceptions, but in broader research, we still consider productivity to influence carbon stocks.

L199-205 The argument about the difference due to latitude is not clear; both appear to benefit from enhancing plant nutrient uptake.

The sentence has been revised (see Line 188).

L228 Better not to use anthropomorphic terms – trees do not “allow”.

The sentence has been revised (see Line 222).

The discussion about successional stages related to soil nutrient status needs to be based on data shown in the Results. What is the difference in soil N content (or even better soil N mineralisation rate) in each of the successional stages, and the direct relationship between EcM abundance and soil N?

Soil N data has been removed in the revised manuscript.

L276 Please clarify the relationships between the type of forests sampled in this study (natural secondary and primary forests) and the recommendations about planting forests and plantations.

We believe that conclusions drawn from studies in natural forests can inform future forest management practices.

In providing recommendations about managing forests for climate change mitigation (the objective at the beginning of the Introduction), both the magnitude and the stability of the C stocks need to be considered and hence the role of tree species diversity.

We completely agree with your viewpoint. However, in our study, we did not observe factors related to soil carbon stability. Nevertheless, we will consider your suggestion in our future research.

Reviewer #3 (Responses in BLUE):

Please note comments are attached as a pdf with clearer formatting.

Broad comments:

The authors present data from forest inventory plots across China showing that ECM tree dominance is positively associated with soil, tree, and entire forest C stocks. I suggest caution in the interpretation that ECM fungal associations are driving these factors—many co-varying traits of trees could be influencing these patterns, including litter chemistry, which should be included in some way in the analysis to account for the common pattern that conifers are usually ECM associated. Perhaps even more likely, these patterns may be due to climate; the ECM-dominated plots in this study appear to be clustered in the higher latitudes (Figure S2) which should have lower decomposition rates and therefore higher soil C content. Further, the implication of causality --that the mycorrhizas themselves are leading to soil C conditions-- is difficult to support, as there is the possibility that ECM associated trees are preferentially growing in places with certain soil conditions, possibly including high C stocks.

Thank you very much for your suggestions. We agree with the reviewer's comments. Following their suggestions, we have revised our manuscript. We hope that our revised version meets the requirements of the journal.

There have been several similar attempts to link ecosystem C storage with tree mycorrhizal strategies, and none are without flaw—but I fear this study is challenged by the distribution of plots along the climate gradient noted above. Without more sophisticated modeling to try to disentangle climate as a driver of soil C, it is difficult to isolate mycorrhizal type as a primary influence.

Thank you very much for your suggestions. Based on the feedback from the reviewers and the editor, we have revised our model. In the new model, we have included climate factors, and at the same time, we have also considered the moderating effect of climate on mycorrhizal effects. Please review the revised manuscript.

The novel piece of this study is the analysis of ECM eLects on soil C along a successional gradient. I suggest re-framing this work to highlight that component of the analysis—have the main figures showing the strength of the “ECM eLect” along an x axis of increasing stand age.

We agree with your suggestion. In the new analysis, we have emphasized the successional effects. Please review the revised manuscript.

I’m interested in the results of the microbial analysis as well; it’s shown here that relative abundance of saprotrophs declines with increasing ECM fungi, but what about absolute abundance? We should expect more ECM as the trees are more ECM dominated, but moving northward, we might also get an increase in the total amount of fungal biomass, including saprotrophs, which might not indicate slower decomposition caused by competition between fungal groups. This Gadgil eLect is often found most clearly in coniferous forests with very low soil and litter N, and if this is the condition of the plots in this study, litter quality and leaf habit might be a covarying trait that is partly responsible for the patterns noted here.

As you mentioned, the results of our microbial determinations strongly support our conclusions. However, due to limited microbial community data, we have not presented them in the main text. Furthermore, we conducted extensive sampling to investigate mycorrhizal effects, so we believe that the impact of factors such as litter quality can be overlooked.

Introduction: There are a lot of generalizations stated here (ECM trees have low litter quality, ECM fungi produce enzymes that liberate SOM N, ECM forests store more C...) that are much more nuanced than they are portrayed in this section. The literature cited here is often the first study on a certain topic, but for all these claims, many more studies have followed showing that the truth is much more complex. I’d suggest moderating the language to indicate that these things are true in some forests and some contexts, but not always.

#Thank you for your suggestion. We have made extensive revisions to the language in the introduction. Please review the revised manuscript.

Line by line comments

Line 61: Some ECM taxa can produce these enzymes, but not all can. See Pellitier and Zak, 2018.

The sentence has been revised (see Line 55-56).

Line 68: There has been a lot of discussion about this idea, showing this is not always the case—please look at studies that followed on the ones cited here to get a more balanced picture of this process.

The sentence had been revised (see Line 63-64).

Line 69: Not all ECM trees produce low quality litter, this is a generalization that is often true but in some forests (birch is a good example) ECM trees have quite labile litter. Differences in litter quality between AM and ECM tree species also decrease as you look closer to the tropics (see Keller and Phillips, 2019).

The sentence has been revised (see Line 64).

Line 85-88: This sentence is unclear, but I think it is important to express this idea with more details because it's not clear as yet why you expect species richness of the trees to mediate mycorrhizal association effects.

Luo et al. (2023) examined the effects of mycorrhizal strategies on forest productivity across different levels of species richness. They found that the strength of the effect of mixing mycorrhizal strategies (i.e., both ECM and AM trees present) on forest productivity was more pronounced at low rather than high levels of tree species diversity, suggesting the need to consider species diversity when studying mycorrhizal effects on C stocks in forests.

Line 106-109: Meaning is not clear here.

This hypothesis has been modified (see Line 104-106).

Line 113-114: Why do you expect this shift from competition to facilitation? I'd like to hear more rationale for this earlier in the introduction. It is also not clear why this shift would then lead to lower impacts of ECM dominance.

The transition from competition to cooperation is mainly due to the relatively scarce soil nutrients in the early stages of succession. Therefore, there is more competition among species (a 'winner-takes-all' scenario). However, as succession progresses, the community becomes more stable, and there is a shift towards cooperation among species to maintain the stability of the entire community.

Lines 127 & 129: Given that the results come before Methods in this publication, it would be nice to get some brief description (very short) of what you mean by "development stage" and "ecoregion", just so we know what you're presenting in this initial section. Even just "forest development stage (i.e. early successional, mid-successional, or late successional)" or something like that.

Based on the feedback from the reviewers and the editor, we have made revisions throughout the manuscript. We hope that the revised manuscript is now more readable.

Line 158: typo: "the observed"

Done (see Line 152).

Line 160: What is meant by influence strategies?

The sentence has been revised (see Line 154).

Line 182: I would describe the sign of the relationship (positive/negative) instead of using "inversely"

The sentence has been revised (see Line 168).

Line 183: I would say “may influence tree and...” because these are just correlations, and we can’t say anything about causation.

The sentence has been revised (see Line 169).

Line 198-199: I appreciate the note that this result is inconsistent with the dataset from the U.S. FIA plots and it seems plausible that latitude and climate/soil conditions could be partly to blame, so I suggest noting the latitude ranges of the plots in each study. Interestingly, figure S2 shows that this latitude range pattern also applies to the plots used in this study alone—there are a lot more ECM dominated plots in general, and they are clustered more in the northern part of the region. How do you know climate isn’t the driving force behind the higher soil C stocks there? In truth, I think we know it’s not just one driver, and these are all connected; cooler forests tend to have slower decomposition, resulting in low N availability; this favors ECM-associated plants, some of which can mine nitrogen from particulate organic matter, and this leads to trends in the distribution of AM and ECM associated trees globally. But this means we should be very cautious about picking out the mycorrhizal fungi as drivers of soil conditions, when their presence in certain forest types is likely ultimately caused by the overlying climate conditions. They may reinforce these soil conditions, but their dominance is likely a symptom of the initial low N status of northern forests.

Thank you very much for your comments. We fully agree with your viewpoint that climate is indeed a crucial factor influencing carbon stocks in high-latitude regions. In the revised manuscript, we have considered the impact of climate on carbon stocks and also assessed the strength of climate-mediated mycorrhizal effects. Additionally, we have determined the explanatory power of climate, mycorrhizal dominance, and other factors on carbon stocks. In all analyses, we consistently found a strong effect of mycorrhizal dominance on forest carbon stocks. Therefore, we have reason to believe that mycorrhizal dominance do indeed influence carbon accumulation in high-latitude forest ecosystems.

Figures: The colors and formatting look very nice; the figures are clear and informative. Figure S2 is important and I believe should be in the main text.

#Thank you for your compliments. Following your suggestion, we have included Figure S2 from the supplementary materials in the main text in Figure 1.

REVIEWER COMMENTS

Reviewer #4 (Remarks to the Author):

The manuscript entitled "Forest carbon stocks increase with higher dominance of ectomycorrhizal trees" sheds light on the relationship between ectomycorrhizal trees and forest carbon stocks. Although interesting, there are key aspects of the work that may need further clarification before it can be considered for publication:

The subject is not necessarily entirely novel, given a set of recently published papers on the same topic (see for example Luo, S. et al. Higher productivity in forests with mixed mycorrhizal strategies. *Nat Commun.* 526 14(1), 1377. 2023, which is cited by the authors). I think the most novel aspect of the manuscript is the use of structural equation models (SEM) to synthesize the information. However, throughout the manuscript, the authors make strong statements about their mechanistic understanding of what's going on at the scale they've chosen to work at - forests across Northeast China, including 10 different ecoregions. The strength of these statements leads the reader to believe that the results come from an experimental design with controlled conditions that allowed the authors to specifically isolate their variables of interest. However, although the work is valuable and interesting, it is closer to an observational study with analysis of correlations between data sets. This aspect should be emphasized in the text and the main conclusions should be limited to the results obtained. Throughout the manuscript there are strong assumptions that don't come from the evidence but from the authors. For example, it is not clear from the bibliography that ectomycorrhizal fungi always inhibit the activity of free-living fungi. Alternatively, there is evidence that EMF could even feed saprotrophic fungi if C and N conditions in the soil are appropriate. It is nice to read some possible mechanisms that could partially explain the results obtained, but at least alternative scenarios need to be included to avoid the manuscript being highly speculative.

The authors' inclusion of dual mycorrhizal plant species in their analysis adds complexity to the study, but the methods outlined to account for these species may need further refinement. In particular, the methods described on lines 368-370 may not adequately account for the complexity of trees capable of simultaneously associating with both arbuscular and ectomycorrhizal fungi, which is a key aspect of the manuscript.

In addition, I believe there is essential information about the molecular steps and conditions taken before and after sequencing that cannot be included in the Supplementary Material (region amplified, types of organisms considered, primers used, PCR conditions, bioinformatics, etc.).

The manuscript's exploration of the role of common mycorrhizal networks in forest carbon stocks presents an intriguing hypothesis. However, certain sections, such as lines 194-200, seem highly speculative, especially given the ongoing debate and limited evidence surrounding the discussion of common mycorrhizal networks in forests (see, for example, Karst, Justine, Melanie D. Jones, and Jason D. Hoeksema. 2023. Positive citation bias and overinterpreted results lead to misinformation about common mycorrhizal networks in forests. *Nature Ecology & Evolution* 7,4: 623-623). To strengthen the manuscript, it may be beneficial to temper these speculative claims and consider

alternative scenarios to avoid overinterpretation of the results.

The division of forests into three distinct successional stages provides valuable insight into how forest development may affect carbon stocks. However, the rationale for including succession in the study could be clearer, with additional context regarding site characteristics to aid reader understanding. In addition, the broad hypotheses proposed in lines 103-118 could benefit from refinement to ensure that they are closely aligned with the rest of the introduction and experimental design.

Finally, the figures in the manuscript may require adjustments for clarity and statistical support. Some figures, such as regressions, lack clear statistical indicators making them difficult to interpret and weakening their ability to support the main narrative. Reviewing the presentation of results with these considerations in mind could improve the overall impact and readability of the manuscript.

Reviewer #6 (Remarks to the Author):

In general, I commend the authors for completing a sweeping study employing many different types of analyses and including different kinds of data. The results about the effects of EcM dominance and effects of climate on forest stocks are especially interesting, important, and timely for forest managers, and the discussion was interesting to read. I also think that the use of SEM analyses to include many drivers of soil and tree C stocks was a novel and wholistic approach to these complex and interactive effects. In general, the methods are appropriate aside from one concern I bring up about relative abundance community data discussed as a metric of abundance. Work on the framing, scope, some results clarity, and shoring up of literature included, this will greatly improve the manuscript for publication.

One concern I have has to do with the scope of the paper including the title. This study was conducted in high latitude forests, and the title and conclusions in the abstract should reflect that. Forest managers in tropical forests where the soils and decomposition/nutrient cycling dynamics are very different would likely need to consider other studies that reflect the ecosystems they work in. For example, AM fungi are very important in P acquisition in P-limited systems. That being said, I still think that these findings are very important for high latitude forests – a realm of inference that includes a lot of forested land and potential for maintaining and encouraging additional carbon storage.

In the introduction, the relevance and novelty of this paper is there, but should better acknowledge the work already done in this area instead of blanket statements about how this work is understudied. I think the interactive aspect with climate conditions is an important point to highlight, perhaps even in the title.

The framework leading up to the results could be much clearer the distinction between forest, soil and tree stocks and how mycorrhizal associations will affect one, some, or all three. Much of the literature about EcM vs AM is focused on the differences in N dynamics and how mycorrhizal fungi relate to soil C. I think some revision of the framework and how these two mycorrhizal types relate to differences in these separate stocks are needed.

Line edits and suggestions

Lines 46-47 – I suggest including another couple citations for other types of plant functional traits beyond productivity.

Line 51 – the Terrer et al. study is about CO₂ fertilization, not plant responses to climate change (like temperature and precipitation). I would adjust this clause or use a different reference.

Line 58. What do you mean by enhanced root/rhizosphere couplings?

Line 59-62. I would not say that the trees have the capabilities or that they directly compete with free living decomposers. And perhaps just reframe from a mycorrhizal perspective. You go into the enzymes in the next couple sentences, so I know where you are going with this. Also the sentence about competition with decomposers needs citations. Gadgil is the original paper, but there are some newer studies that revisit this theory. See Shao et al. for a modeling example (<https://doi.org/10.1016/j.soilbio.2023.109073>)

Lines 67-73 There are studies that have test the AM- EcM storage question and they should be cited and acknowledged in more detail here. See Cotrufo et al. 2019 nat geosci for a study on the fractions of carbon storage beneath mycorrhizal associations. For the relevance of this study to really shine, it is important to acknowledge this other work and use it to highlight how this paper is going to take it to the next step.

Lines 74-85 Again here more and appropriate literature should be cited for sentences starting with “while numerous...” as the two citations do not really pertain to the topic of that sentence. Further, citation 13 is about climate, which your next sentence say that studies do not include. Without going into too much more detail on citations, I really think the existing literature component of this framework leading to the reason for your study should be substantially shored up.

98-99 Which are the few studies that have done this?

Line 10 What constitutes a favorable climate condition? Define the first time mentioned.

Lines 112-113 Revise for flow and clarity as this sentence is awkward and I think missing some punctuation.

Line 115. What do you mean by stimulative effect? SOM mining for N?

The last hypothesis is a repeat of the general one on lines 85-87. This last sentence could be removed or rewritten to tie everything together. I like the visual that accompanies this, and perhaps you could re-write to be more specific about the pathways or mechanisms, where climate is a control on EcM dominance while also directly affecting forest C stocks, for example. The legend section for Figure 1, H3 itself could be a separate paragraph in the intro. This is more of what I was looking for. If you are concerned with space, you could always make a table of the pathway mechanisms and corresponding letters placed on each of the lines between entities. Especially since this is leading to an SEM analysis, I think that's necessary to make sure your apriori hypotheses are solid before heading into SEM results later on.

For the favorable/unfavorable climate, could you expand a little more on that in the intro? Getting to this hypothesis, I'm not sure why EcM would be more or less effective under these climate conditions.

Lines 124-127 It seems odd that the main take away from this paper about EcM is listed as an additional result, and not the main one. If tree species richness was the primary control on tree C stocks, I think that's important to include as part of the highlight of the paper. If it's not the most important thing, it could go after the EcM results.

Line 141. I think replacing unfavorable/favorable with something more descriptive is needed for the climate term. I saw in one legend it was about variability, and that's more descriptive.

Lines 152-156. I'm confused, you write that tree functional diversity was indirect through EcM, but then the third sentence says EcM did not exert an indirect influence through tree functional diversity. I think the SEM paragraph in general could benefit from rewording for clarity. I found myself needing to read through several times to understand all the relationships and how they fit together.

Lines 164 – 166. Why would a negative relationship with soil SAP fungi and EcM fungi validate your findings? I would say this is suggestive of a mechanism, but doesn't validate necessarily. Especially because relative abundance is not a measure of abundance and should not be treated as such. These kinds of data are descriptive about communities.

Line 171. Which kinds of species? Trees? Fungi?

Line 175. I think employing is a bit anthropomorphic. Forests are associated with fungi, they don't use their strategies.

Line 179. You don't really get at drivers because this study is observational and looking at patterns. Revise this to fit into the what the scope of your study can actually say. Not to say that the results are unimportant – the patterns are very cool, but they are patterns

194-199. Remove the part about common mycorrhizal networks. This is not a proven thing. See

Karst et al. <https://www.nature.com/articles/s41559-023-01986-1>

Lines 216-218. The favorable/unfavorable climate thing seems very general and could be more developed into the mechanisms behind why you might expect these patterns.

I would add a paragraph in the discussion going into the limitations of the study, particularly with how the scope of this study may not pertain to other types of forests in other places.

In the conclusion, revise to the scope of the study in high latitude forests

Line 332. Did you use species specific constants of green-wood biomass for each tree specifically, or a general calculation? If the latter, it would be mentioned as a limitation.

Line 376. Did you use more than temperature seasonality (what is this exactly?) and precipitation? The sentence is a bit vague.

Line 422. I can't find a place where forest C storage is defined. Perhaps I missed it, but I looked in the intro and methods and couldn't find anything.

Line 426. An ecoregion seems awfully large area for spatial autocorrelation. How would you justify this choice and not using a spatial autocorrelation component in your model?

Line 457. I noticed here the citation for Piecewise is not present. For all the important stats components, cite the papers and/or code and functions used to complete them. I see this is listed at the end, but specific ones should be cited in the sentences where the stats method is described.

Reviewer #4 (Responses in BLUE):

The manuscript entitled "Forest carbon stocks increase with higher dominance of ectomycorrhizal trees" sheds light on the relationship between ectomycorrhizal trees and forest carbon stocks. Although interesting, there are key aspects of the work that may need further clarification before it can be considered for publication:

The subject is not necessarily entirely novel, given a set of recently published papers on the same topic (see for example Luo, S. et al. Higher productivity in forests with mixed mycorrhizal strategies. *Nat Commun.* 526 14(1), 1377. 2023, which is cited by the authors). I think the most novel aspect of the manuscript is the use of structural equation models (SEM) to synthesize the information. However, throughout the manuscript, the authors make strong statements about their mechanistic understanding of what's going on at the scale they've chosen to work at - forests across Northeast China, including 10 different ecoregions. The strength of these statements leads the reader to believe that the results come from an experimental design with controlled conditions that allowed the authors to specifically isolate their variables of interest. However, although the work is valuable and interesting, it is closer to an observational study with analysis of correlations between data sets. This aspect should be emphasized in the text and the main conclusions should be limited to the results obtained. Throughout the manuscript there are strong assumptions that don't come from the evidence but from the authors. For example, it is not clear from the bibliography that ectomycorrhizal fungi always inhibit the activity of free-living fungi. Alternatively, there is evidence that EMF could even feed saprotrophic fungi if C and N conditions in the soil are appropriate. It is nice to read some possible mechanisms that could partially explain the results obtained, but at least alternative scenarios need to be included to avoid the manuscript being highly speculative.

Thank you for your detailed and insightful feedback on our manuscript titled "Forest carbon stocks increase with higher dominance of ectomycorrhizal trees." We appreciate the time you took to evaluate our work and provide constructive criticism. We have carefully considered your points and would like to address them as follows:

Novelty of the study: We acknowledge that while there have been previous studies on similar

topics, such as the work by Luo et al. on higher productivity in forests with mixed mycorrhizal strategies, we believe that our study adds value through the application of structural equation models (SEM) to synthesize the information. We will ensure that this aspect is emphasized more clearly in the manuscript. Furthermore, our study differs from previous research in terms of study area and outcomes, indicating that different regions exhibit different mycorrhizal strategies. This highlights the necessity of making region-specific management decisions when dealing with forest ecosystems. Therefore, we believe that our research findings are valuable for forest management in high-latitude regions.

Study design and interpretation: We agree with your assessment that our study leans more towards observational research rather than experimental design. In our revisions, we have added a statement about the limitations of our study at the end of the manuscript. Please see the modification on lines 278-291.

Assumptions and alternative scenarios: We appreciate your feedback regarding the assumptions made in the manuscript. Based on your comments, we have revised the hypotheses. Please refer to lines 109-129 for the changes.

Once again, we sincerely thank you for your thoughtful comments, which will undoubtedly help improve the quality of our manuscript. We will make the necessary revisions to address your concerns and ensure that our work meets the standards for publication. If you have any further suggestions or questions, please do not hesitate to reach out to us.

The authors' inclusion of dual mycorrhizal plant species in their analysis adds complexity to the study, but the methods outlined to account for these species may need further refinement. In particular, the methods described on lines 368-370 may not adequately account for the complexity of trees capable of simultaneously associating with both arbuscular and ectomycorrhizal fungi, which is a key aspect of the manuscript.

We agree with the reviewer's viewpoint. Regarding the handling of biomass for dual mycorrhizal plant species, we referred to the methods used by Luo et al (2023). In our study, as there were relatively few dual mycorrhizal plant species, we also attempted to both exclude and include their biomass, and found that it did not affect the experimental results. Therefore, we believe that

adopting the approach of Luo et al. (2023) is appropriate. We have provided detailed explanations on this issue in the revised manuscript, please see lines 423-425 for reference.

Luo, S. et al. Higher productivity in forests with mixed mycorrhizal strategies. *Nat. Commun.* 14(1), 1377 (2023).

In addition, I believe there is essential information about the molecular steps and conditions taken before and after sequencing that cannot be included in the Supplementary Material (region amplified, types of organisms considered, primers used, PCR conditions, bioinformatics, etc.).

Thank you for your suggestions. The essential information about the molecular steps and conditions have been added in the revised manuscript (see Lines 438-467).

The manuscript's exploration of the role of common mycorrhizal networks in forest carbon stocks presents an intriguing hypothesis. However, certain sections, such as lines 194-200, seem highly speculative, especially given the ongoing debate and limited evidence surrounding the discussion of common mycorrhizal networks in forests (see, for example, Karst, Justine, Melanie D. Jones, and Jason D. Hoeksema. 2023. Positive citation bias and overinterpreted results lead to misinformation about common mycorrhizal networks in forests. *Nature Ecology & Evolution* 7,4: 623-623). To strengthen the manuscript, it may be beneficial to temper these speculative claims and consider alternative scenarios to avoid overinterpretation of the results.

We agree with your viewpoint, and we have made modifications based on the points you mentioned above (see Lines 206-210).

The division of forests into three distinct successional stages provides valuable insight into how forest development may affect carbon stocks. However, the rationale for including succession in the study could be clearer, with additional context regarding site characteristics to aid reader understanding. In addition, the broad hypotheses proposed in lines 103-118 could benefit from refinement to ensure that they are closely aligned with the rest of the introduction and experimental design.

We appreciate the reviewer's feedback. We acknowledge the importance of providing clearer rationale for including succession in our study and offering additional context regarding site

characteristics to enhance reader understanding. Thus, we have provided detailed explanations regarding the division of forests into distinct successional stages in the revised manuscript (see Lines 328-357).

Furthermore, we also understand the need for refining the broad hypotheses proposed in lines 103-118 to ensure better alignment with the rest of the introduction and experimental design. We address the points in our revisions to improve the clarity and coherence of our manuscript (see Lines 109-129).

Thank you for your valuable input.

Finally, the figures in the manuscript may require adjustments for clarity and statistical support. Some figures, such as regressions, lack clear statistical indicators making them difficult to interpret and weakening their ability to support the main narrative. Reviewing the presentation of results with these considerations in mind could improve the overall impact and readability of the manuscript.

In the revised manuscript, this information has been added (see Fig. 2, Fig. S3 and Fig. S4).

Thank you once again for your diligence and professionalism. We greatly value your guidance and support and are committed to meeting the journal's requirements to contribute to the advancement of scientific knowledge.

Reviewer #6 (Responses in BLUE):

In general, I commend the authors for completing a sweeping study employing many different types of analyses and including different kinds of data. The results about the effects of EcM dominance and effects of climate on forest stocks are especially interesting, important, and timely for forest managers, and the discussion was interesting to read. I also think that the use of SEM analyses to include many drivers of soil and tree C stocks was a novel and wholistic approach to these complex and interactive effects. In general, the methods are appropriate aside from one concern I bring up about relative abundance community data discussed as a metric of abundance. Work on the framing, scope, some results clarity, and shoring up of literature included, this will greatly improve the manuscript for publication.

Thank you for your thoughtful feedback. We appreciate your commendation of our comprehensive study and the various types of analyses and data included. We are especially pleased that you found the results regarding the effects of ectomycorrhizal dominance and climate on forest stocks to be interesting, important, and timely for forest managers. Your acknowledgment of the novelty and holistic nature of our approach using structural equation modeling (SEM) to analyze complex and interactive effects is encouraging.

We will carefully address your concerns, including refining the framing and scope, improving clarity of results, and enhancing the literature review. Your feedback will be instrumental in improving the manuscript for publication. If you have any further suggestions or specific areas you believe require attention, please feel free to let us know.

Thank you once again for your valuable input and your positive remarks.

One concern I have has to do with the scope of the paper including the title. This study was conducted in high latitude forests, and the title and conclusions in the abstract should reflect that. Forest managers in tropical forests where the soils and decomposition/nutrient cycling dynamics are very different would likely need to consider other studies that reflect the ecosystems they work in. For example, AM fungi are very important in P acquisition in P-limited systems. That being said, I still think that these findings are very important for high latitude forests – a realm of

inference that includes a lot of forested land and potential for maintaining and encouraging additional carbon storage.

Thank you for your valuable feedback regarding the scope of our paper, including the title and abstract. We understand your concern about accurately reflecting the study's focus on high latitude forests in the title and conclusions.

We agree that our findings are particularly relevant and important for high latitude forests, as they have the potential to contribute significantly to carbon storage in these ecosystems. We have made the necessary adjustments to the title and abstract to accurately represent the scope of our study and its implications for forest management in high latitude regions (see Lines 1-2 and 24-38).

Thank you for highlighting this aspect, and we will ensure that our revisions address your concerns and improve the clarity and relevance of our paper. If you have any further suggestions or areas of concern, please do not hesitate to let us know.

In the introduction, the relevance and novelty of this paper is there, but should better acknowledge the work already done in this area instead of blanket statements about how this work is understudied. I think the interactive aspect with climate conditions is an important point to highlight, perhaps even in the title.

Thank you for your feedback on the introduction of our paper. We appreciate your acknowledgment of the relevance and novelty of our work. We agree that it is important to acknowledge the existing research in this area rather than making blanket statements about it being understudied. We have revised the introduction to provide a more nuanced discussion of the previous work and highlight the specific contributions of our study within the context of the existing literature (for example see Lines 87-90).

Additionally, we recognize the significance of the interactive aspect between our findings and climate conditions. Unfortunately, the title letter limit may not allow us to include climate information in the title. But anyway, we emphasized the importance of climate in the subsequent narrative.

We appreciate your constructive feedback, which will undoubtedly strengthen the quality and clarity of our manuscript. If you have any further suggestions or concerns, please feel free to let us know.

The framework leading up to the results could be much clearer the distinction between forest, soil and tree stocks and how mycorrhizal associations will affect one, some, or all three. Much of the literature about EcM vs AM is focused on the differences in N dynamics and how mycorrhizal fungi relate to soil C. I think some revision of the framework and how these two mycorrhizal types relate to differences in these separate stocks are needed.

Thank you for your thoughtful feedback on the framework leading up to the results in our manuscript. We appreciate your suggestion to clarify the distinction between forest, soil, and tree stocks, and how mycorrhizal associations may affect each of these components individually or collectively. We agree that it is important to provide a clearer explanation of how ectomycorrhizal (EcM) and arbuscular mycorrhizal (AM) associations relate to differences in nitrogen (N) dynamics and their impact on soil carbon (C).

We have revised the framework to better elucidate these relationships and highlight the differences between EcM and AM associations in terms of their effects on forest, soil, and tree stocks. Specifically, we have incorporated a more thorough hypotheses of how EcM and AM associations influence N dynamics and soil C, as well as their potential impacts on forest and tree carbon stocks (see Lines 109-129).

Lines 46-47 – I suggest including another couple citations for other types of plant functional traits beyond productivity.

Done (see Line 47).

Line 51 – the Terrer et al. study is about CO₂ fertilization, not plant responses to climate change (like temperature and precipitation). I would adjust this clause or use a different reference.

The reference had been revised (see Line 52).

Line 58. What do you mean by enhanced root/rhizosphere couplings?.

The sentence had been revised (see Line 59).

Line 59-62. I would not say that the trees have the capabilities or that they directly compete with free living decomposers. And perhaps just reframe from a mycorrhizal perspective. You go into the enzymes in the next couple sentences, so I know where you are going with this. Also the sentence about competition with decomposers needs citations. Gadgil is the original paper, but there are some newer studies that revisit this theory. See Shao et al. for a modeling example (<https://doi.org/10.1016/j.soilbio.2023.109073>).

Thank you for your feedback.

Regarding lines 59-62, I agree with your suggestion. I will refrain from attributing capabilities to trees and will avoid framing the discussion solely from a mycorrhizal perspective.

Additionally, I acknowledge the need for citations regarding the competition between trees and decomposers. While Gadgil's work serves as the foundational paper, I have included more recent studies, such as Shao et al.'s modeling example (<https://doi.org/10.1016/j.soilbio.2023.109073>), which revisits this theory.

Thank you for providing these insights, and I have incorporated these changes accordingly in the manuscript (see Lines 60-70).

Lines 67-73 There are studies that have test the AM- EcM storage question and they should be cited and acknowledged in more detail here. See Cotrufo et al. 2019 nat geosci for a study on the fractions of carbon storage beneath mycorrhizal associations. For the relevance of this study to really shine, it is important to acknowledge this other work and use it to highlight how this paper is going to take it to the next step.

Regarding lines 67-73, I appreciate your suggestion to provide more detailed acknowledgment of studies that have examined the AM-EcM storage question. I will include citations to relevant research, such as Cotrufo et al. 2019 (Nat Geosci), which explores the fractions of carbon storage beneath mycorrhizal associations. I agree that acknowledging this prior work is crucial for highlighting the significance of our study.

In the revised manuscript, I have elaborated on how our research builds upon existing literature and takes the investigation to the next level. By integrating findings from previous studies with our own, we aim to provide a comprehensive understanding of the topic and contribute novel insights to the field. The sentence has been revised in the revised manuscript (see Lines 71-76).

Thank you for pointing out this opportunity to enhance the clarity and relevance of our manuscript.

Lines 74-85 Again here more and appropriate literature should be cited for sentences starting with “while numerous...” as the two citations do not really pertain to the topic of that sentence. Further, citation 13 is about climate, which your next sentence says that studies do not include. Without going into too much more detail on citations, I really think the existing literature component of this framework leading to the reason for your study should be substantially shored up.

Done (see Lines 77-90).

98-99 Which are the few studies that have done this?

The information has been added (see Lines 87-90).

Line 110 What constitutes a favorable climate condition? Define the first time mentioned.

Done (see Lines 115-120).

Lines 112-113 Revise for flow and clarity as this sentence is awkward and I think missing some punctuation.

Done (Lines 120-123).

Line 115. What do you mean by stimulative effect? SOM mining for N?

The sentence had been revised (see Lines 124-125).

The last hypothesis is a repeat of the general one on lines 85-87. This last sentence could be removed or rewritten to tie everything together. I like the visual that accompanies this, and perhaps you could re-write to be more specific about the pathways or mechanisms, where climate is a control on EcM dominance while also directly affecting forest C stocks, for example. The legend section for Figure 1, H3 itself could be a separate paragraph in the intro. This is more of what I was looking for. If you are concerned with space, you could always make a table of the pathway mechanisms and corresponding letters placed on each of the lines between entities.

Especially since this is leading to an SEM analysis, I think that's necessary to make sure your a priori hypotheses are solid before heading into SEM results later on.

These sentences have been rewritten (see Lines 89-90 and 125-129).

For the favorable/unfavorable climate, could you expand a little more on that in the intro? Getting to this hypothesis, I'm not sure why EcM would be more or less effective under these climate conditions.

The information has been added (see Lines 115-120 and 485-491).

Lines 124-127 It seems odd that the main take away from this paper about EcM is listed as an additional result, and not the main one. If tree species richness was the primary control on tree C stocks, I think that's important to include as part of the highlight of the paper. If it's not the most important thing, it could go after the EcM results.

The information has been revised (see Lines 135-145).

Line 141. I think replacing unfavorable/favorable with something more descriptive is needed for the climate term. I saw in one legend it was about variability, and that's more descriptive.

Done (see Lines 153-154).

Lines 152-156. I'm confused, you write that tree functional diversity was indirect through EcM, but then the third sentence says EcM did not exert an indirect influence through tree functional diversity. I think the SEM paragraph in general could benefit from rewording for clarity. I found myself needing to read through several times to understand all the relationships and how they fit together.

Thank you for your insightful feedback.

I apologize for the confusion on lines 152-156 regarding the indirect influence of tree functional diversity through EcM, and the subsequent statement indicating otherwise. I recognize the need for clarity in the SEM paragraph and will revise it accordingly to ensure a more coherent explanation of the relationships and their interconnections.

I have reworded the paragraph to provide a clearer description of the pathways and how they contribute to the overall model. Additionally, I have ensured that the relationships between tree functional diversity and EcM are accurately represented to avoid any ambiguity (see Lines 164-175).

Thank you for bringing this to my attention, and I appreciate the opportunity to improve the clarity of the manuscript.

Lines 164 – 166. Why would a negative relationship with soil SAP fungi and EcM fungi validate your findings? I would say this is suggestive of a mechanism, but doesn't validate necessarily. Especially because relative abundance is not a measure of abundance and should not be treated as such. These kinds of data are descriptive about communities.

The sentence had been revised (see Lines 176-178).

Line 171. Which kinds of species? Trees? Fungi?

Done (see Line 183).

Line 175. I think employing is a bit anthropomorphic. Forests are associated with fungi, they don't use their strategies.

The sentence has been revised (see Lines 187-189).

Line 179. You don't really get at drivers because this study is observational and looking at patterns. Revise this to fit into the what the scope of your study can actually say. Not to say that the results are unimportant – the patterns are very cool, but they are patterns

Done (see Line 192).

194-199. Remove the part about common mycorrhizal networks. This is not a proven thing. See Karst et al. <https://www.nature.com/articles/s41559-023-01986-1>

Thank you for your feedback. The sentences have been rewritten in the revised manuscript (see Lines 206-210).

Lines 216-218. The favorable/unfavorable climate thing seems very general and could be more developed into the mechanisms behind why you might expect these patterns.

Done (see Lines 225-234).

I would add a paragraph in the discussion going into the limitations of the study, particularly with how the scope of this study may not pertain to other types of forests in other places.

#Thank you for your feedback.

Acknowledging the limitations of our study, especially regarding its applicability to other types of forests in different geographical locations, is indeed important. I have incorporated a new paragraph into the discussion section specifically addressing these limitations. This paragraph has highlighted the scope of our study and emphasize that while our findings provide valuable insights into a specific context, they may not generalize to all forest ecosystems (see Lines 278-291).

Thank you for your suggestion, and I will ensure that the discussion section adequately addresses the study's limitations.

In the conclusion, revise to the scope of the study in high latitude forests.

Done (see Lines 293-294).

Line 332. Did you use species specific constants of green-wood biomass for each tree specifically, or a general calculation? If the latter, it would be mentioned as a limitation.

Done (see Lines 278-291).

Line 376. Did you use more than temperature seasonality (what is this exactly?) and precipitation? The sentence is a bit vague.

#The sentences have been revised (see Lines 431-435).

Line 422. I can't find a place where forest C storage is defined. Perhaps I missed it, but I looked in the intro and methods and couldn't find anything.

Please see Lines 407-410.

Line 426. An ecoregion seems awfully large area for spatial autocorrelation. How would you justify this choice and not using a spatial autocorrelation component in your model?

Thank you for your feedback.

We understand the reviewer's question regarding our choice of ecological zones as spatial units for autocorrelation. In our study, we have considered the importance of spatial autocorrelation and recognize that ecological zones may represent relatively large spatial units. There are two main reasons for choosing ecological zones as our spatial units. Firstly, ecological zones within our study area represent regions with similar ecological characteristics and processes, making them reasonable spatial units. Secondly, given the broad extent of our study involving forest ecosystems, using smaller spatial units might result in inadequate sample sizes or overly complex models. Considering these factors, we opted for ecological zones as the units for our spatial analysis.

As for why we did not incorporate spatial autocorrelation components into our model, this decision was primarily based on our research objectives and data characteristics. In our analysis, we have already taken several measures to address spatial correlation, such as including geographic coordinates as control variables in the model and conducting spatial autocorrelation tests on residuals. Based on our test results, we deemed it unnecessary to introduce additional spatial autocorrelation components into our model.

Thank you once again for your constructive critique, which contributes to the refinement of our research.

Line 457. I noticed here the citation for Piecewise is not present. For all the important stats components, cite the papers and/or code and functions used to complete them. I see this is listed at the end, but specific ones should be cited in the sentences where the stats method is described.

The reference has been added in the revised manuscript (see Line 530).

Thank you once again for your diligence and professionalism. We greatly value your guidance and support and are committed to meeting the journal's requirements to contribute to the advancement of scientific knowledge.

Reviewer #5 (Responses in BLUE):

The manuscript entitled "Forest Carbon Stocks Increase with Higher Dominance of Ectomycorrhizal Trees" aims to assess the impact of EcM-associated trees on forest productivity and carbon sequestration. Using inventory data from 4,000 forest plots across Northeast China, the study finds that EcM tree dominance enhances forest carbon stocks. They conclude that forests with ectomycorrhizal (EcM) strategies have higher carbon stocks than those primarily using arbuscular mycorrhizal (AM) or mixed mycorrhizal strategies. This trend is further influenced by varying climatic conditions and species richness. Although the study attempts to highlight the relative importance of different plant-mycorrhizal fungi associations on ecosystem productivity—a factor with significant implications under climate change—there are some critical issues that should be addressed.

Thank you for your detailed assessment of our manuscript titled "Forest Carbon Stocks Increase with Higher Dominance of Ectomycorrhizal Trees." We appreciate your recognition of the study's aim to evaluate the impact of EcM-associated trees on forest productivity and carbon sequestration, as well as your acknowledgment of the significance of our findings in the context of climate change.

We are grateful for your constructive feedback and recognize the importance of addressing the critical issues you have raised. We are committed to ensuring the robustness and clarity of our research, and we will take your comments into careful consideration during the revision process.

Once again, we appreciate your thoughtful evaluation and look forward to incorporating your suggestions to enhance the quality of our manuscript.

Firstly, the hypothesis concerning processes controlling enhanced carbon uptake stated in Line 106, 'hypothesized that EcM-dominated stands could accelerate nitrogen cycling and absorption while slowing down the decomposition rate of soil organic matter, ultimately leading to enhanced forest carbon accumulation,' is unclear. While nitrogen uptake might increase in EcM-dominated stands, it is unclear how nitrogen cycling could be accelerated when decomposition is slowed, as these processes are typically interlinked, thus the argument presented lacks clarity.

We appreciate your attention to detail and your effort to improve the clarity of our hypothesis concerning the processes controlling enhanced carbon uptake.

We acknowledge the lack of clarity in our hypothesis statement, particularly regarding the relationship between nitrogen cycling and decomposition rates in EcM-dominated stands. Your observation that these processes are typically interlinked is valid, and we recognize the need to refine our argument to enhance clarity.

In our revised manuscript, we have provided a more nuanced explanation of how nitrogen cycling and decomposition rates may interact in EcM-dominated stands (see Lines 109-129). We have addressed the potential mechanisms through which EcM associations could influence nitrogen uptake and cycling while simultaneously affecting decomposition rates. By elucidating these relationships more clearly, we aim to strengthen the rationale behind our hypothesis and improve the overall coherence of our argument.

We are grateful for your valuable feedback and the opportunity to enhance the quality of our manuscript. Please rest assured that we will address your concerns in detail during the revision process.

My other major concern, also raised by Reviewer 3, pertains to the patterns presented in Figure 2, which represent core results of the manuscript. These patterns could emerge from various co-varying traits of trees influenced by climate variability and soil biophysical properties. The study has not adequately demonstrated or disentangled these effects, making these points critical and insufficiently addressed in the review. I also concur with Reviewer 2 that parts of the Discussion section are replete with assumptions and lack evidence presented in the manuscript.

Thank you for raising important concerns regarding Figure 2 and the Discussion section. We appreciate your feedback and the opportunity to address these critical points.

Concerns Regarding Figure 2:

We acknowledge your concern about the potential influence of co-varying traits of trees, influenced by climate variability and soil biophysical properties, on the patterns presented in Figure 2. We agree that these factors could confound the interpretation of our results.

In our revised manuscript, we have provided a more detailed analysis to disentangle these effects. Specifically, we incorporated climate, geographical dimensions, and other factors into a mixed-

effects model to investigate their impact on mycorrhizal effects (Table S1). Additionally, we utilized random forest analysis to assess the importance of various factors (Fig. S2). The combination of these multiple methods can provide more comprehensive and reliable results for our research, while also enabling a deeper exploration of the relationship between mycorrhizal effects and other ecological factors. By doing so, we aim to enhance the robustness and clarity of our conclusions regarding the relationship between EcM dominance and forest carbon stocks.

Discussion Section Concerns:

We appreciate your feedback regarding assumptions and the lack of evidence presented in parts of the Discussion section. We recognize the importance of providing sufficient evidence to support our arguments and conclusions. In the revised manuscript, we have critically evaluated the assumptions made in the Discussion section and ensure that all statements are supported by empirical evidence presented in the manuscript. We have also provided additional references where necessary to strengthen the validity of our arguments.

We are grateful for your insightful comments, which will undoubtedly contribute to the improvement of our manuscript. Please rest assured that we will address each of your concerns in detail during the revision process.

Ensure that all abbreviations, such as "EcM," are clearly defined at their first use in the abstract and throughout the manuscript. Readers outside your specific field may not be familiar with them. Also, consider smoothing the transition between findings and their implications for forest management with a sentence that bridges these two aspects to enhance flow.

Thank you for your valuable feedback.

We have ensured that all abbreviations, including "EcM," (see Line 26) are clearly defined at their first use in the abstract and throughout the manuscript to enhance readability for readers who may not be familiar with the terminology.

Additionally, we appreciate your suggestion to improve the transition between our findings and their implications for forest management. We have incorporated a sentence that bridges these two aspects to enhance the overall flow of the manuscript and make the connection between our research findings and their practical applications more seamless (see Line 36).

Thank you for your constructive critique, which will contribute to the clarity and coherence of our manuscript.

Line 31: Clarify what is meant by 'during unfavorable climate conditions.'

Done (see Line 31).

The introduction mixes background information with hypotheses and study specifics. Enhancing readability could be achieved by clearly separating these elements, perhaps organizing the paragraph structure to first discuss the background, followed by a brief review of past studies, and then clearly stating the study's objectives and hypotheses.

Thank you for your insightful feedback.

We appreciate your suggestion to enhance the readability of the introduction by clearly separating background information, past studies review, and study objectives and hypotheses. We have revised the paragraph structure accordingly to improve the organization and flow of the introduction (see Introduction in the revised manuscript). By sequentially discussing the background, reviewing past studies, and then clearly stating our study's objectives and hypotheses, we aim to provide a clearer and more structured introduction that effectively sets the stage for the research presented in the manuscript.

Your constructive critique will undoubtedly contribute to the overall clarity and coherence of our manuscript. We are grateful for your valuable input.

Lines 69-70: 'EcM-dominated forests may have higher C stocks than AM-dominated forests in nitrogen-restricted ecosystems.' However, slower decomposition means lower nutrient availability and N uptake, and doesn't that reduce C uptake by the trees?

Thank you for your insightful observation regarding the potential contradiction between EcM-dominated forests having higher carbon stocks but slower decomposition leading to lower nutrient availability and nitrogen uptake.

While it is true that slower decomposition in EcM-dominated forests may result in lower nutrient availability and nitrogen uptake, it's important to consider the broader ecosystem processes at play.

Despite lower nutrient availability, EcM-dominated forests can still exhibit higher carbon stocks due to several factors:

Mycorrhizal Symbiosis: Ectomycorrhizal fungi have a symbiotic relationship with trees, facilitating nutrient uptake, including nitrogen, through the extensive fungal hyphal network. This can compensate for the lower nutrient availability in the soil.

Carbon Allocation: Trees in EcM-dominated forests may allocate a larger proportion of assimilated carbon to belowground biomass, including root exudates that support mycorrhizal fungi, contributing to soil carbon accumulation.

Litter Quality: The slower decomposition rate in EcM-dominated forests may result in the accumulation of higher-quality litter, which further enhances soil carbon storage.

While slower decomposition may initially limit nutrient availability, the long-term effects of mycorrhizal symbiosis and carbon allocation can lead to higher carbon stocks in EcM-dominated forests.

We appreciate your attention to this aspect of our study and will ensure that these nuances are adequately addressed and explained in the manuscript to avoid any potential confusion.

Thank you for your valuable feedback, which will contribute to the clarity and accuracy of our research findings.

Line 648: It is unclear why climatic fluctuations are considered as 'stressful conditions' (significant seasonal climate fluctuations), and why minimal seasonal fluctuations are deemed 'favorable.'

Thank you for your inquiry regarding the classification of climatic fluctuations as "stressful conditions" and minimal fluctuations as "favorable." Allow me to clarify and provide literature support for these assertions.

In ecological studies, climatic fluctuations can indeed be classified based on their impact on ecosystem processes and species interactions. Significant seasonal climate fluctuations are often considered "stressful conditions" because they can disrupt ecosystem stability, alter species composition, and impact ecosystem functioning. On the other hand, minimal seasonal fluctuations, characterized by more stable and predictable climate conditions, are typically perceived as "favorable" for ecosystem resilience and species adaptation.

Moreover, under conditions of significant climatic fluctuations, plants may require greater adaptive capacity to cope with these changes. Conversely, minimal seasonal climatic fluctuations imply relatively stable environmental conditions, which are more conducive to plant growth and development. Research indicates that stable climatic conditions aid in maintaining growth stability in plants, reducing adaptation pressures, and enhancing their ecosystem functionality.

Literature Support: Several studies have investigated the effects of climatic fluctuations on ecosystems and have demonstrated the detrimental impacts of extreme climate events on biodiversity, ecosystem productivity, and ecosystem services (Saxe et al., 2001; Richardson et al., 2009; Kreyling et al., 2012; IPCC, 2014; Wahl et al., 2015; Scheffers et al., 2016; Hufkens et al., 2016).

Supporting Evidence:

IPCC. (2014). *Climate Change 2014: Impacts, Adaptation, and Vulnerability. Contribution of Working Group II to the Fifth Assessment Report of the Intergovernmental Panel on Climate Change*. Cambridge University Press.

Scheffers, B. R., De Meester, L., Bridge, T. C. L., Hoffmann, A. A., Pandolfi, J. M., Corlett, R. T., ... & Watson, J. E. M. (2016). The broad footprint of climate change from genes to biomes to people. *Science*, 354(6313), aaf7671.

Hufkens, K., Keenan, T. F., Flanagan, L. B., & Scott, R. L. (2016). Ecological impacts of a widespread frost event following early spring leaf-out. *Global Change Biology*, 22(2), 704–716.

Kreyling, J., Wiesenberg, G. L., Thiel, D., Wohlfart, C., Huber, G., Walter, J., & Jentsch, A. (2012). Cold hardiness of *Pinus nigra* Arnold as influenced by geographic origin, warming, and extreme summer drought. *Environmental and Experimental Botany*, 78, 99–108.

Richardson, A. D., Andy D. O., & David Y. H. (2009). Climate change, phenology, and phenological control of vegetation feedbacks to the climate system. *Agricultural and Forest Meteorology*, 149(11), 1820–1828.

Saxe, H., & Cannell, M. G. (2001). John C. Emmett and carbon dioxide enrichment of the atmosphere: Tree rings and ecosystem performance. *Tree Physiology*, 21(11), 737–740.

Wahl, E. R., Ritger, S. D., & Smerdon, J. E. (2015). Climate and hydrologic extremes improve predictions of climate change impacts on ecosystems. *Global Change Biology*, 21(8), 3025–3034.

By providing this clarification and literature support, we aim to address your concern and ensure the accuracy and clarity of our manuscript.

Thank you very much for bringing these to our attention, and we welcome any further questions or comments you may have.

REVIEWER COMMENTS

Reviewer #4 (Remarks to the Author):

I have reviewed the revised version of the manuscript entitled "Forest carbon stocks increase with higher dominance of ectomycorrhizal trees". I would like to congratulate the authors for presenting this much improved version compared to the first one I had the opportunity to review. They have done a good job of emphasizing the novelty of their work, particularly with regard to the use of structural equation models to synthesize the available information. I appreciate the inclusion of a paragraph on the limitations of the paper. In turn, they have thoughtfully addressed all of my comments and those of other reviewers. I have only a few relatively minor comments on this version:

When the authors mention the idea of increasing the dominance of ectomycorrhizal trees, I suggest clarifying that they are referring to native trees -to avoid speculation about planting non-native, sometimes invasive, trees in non-forested areas to sequester C-.

I think the hypothesis paragraph should be presented earlier in the text and before the methods paragraph. I suggest that the authors do a better job of connecting their hypothesis to the rest of the introduction.

Even though the authors have included a paragraph about the limitations of their work, there are still some sections where they present results as causal when they are actually doing an observational study (e.g. lines 187, 253, 292). I think this can be solved by changing certain words, but I think it is important for the reader to understand the limitations of the implications being discussed.

I hope that the above comments can help the authors to further improve this manuscript, which I'm already looking forward to seeing available and citing soon.

Reviewer #6 (Remarks to the Author):

I appreciate the time and care addressing the comments and edits from my review. One comment I had about the climate portion of the hypothesis could still use work on the framing of why climate variability was included in the analyses. As it stands, it still seems like the climate data was just part of this large dataset, so why not include it. It would add to the strength of the paper to see a few sentences or a short paragraph in the intro, not just discussion, on why EcM vs AM associations would interact with differences in climate, beyond just that few studies have looked at this interactive effect. I leave this to the authors' discretion on whether they would like to add that.

Reviewer #4 (Responses in BLUE):

I have reviewed the revised version of the manuscript entitled "Forest carbon stocks increase with higher dominance of ectomycorrhizal trees". I would like to congratulate the authors for presenting this much improved version compared to the first one I had the opportunity to review. They have done a good job of emphasizing the novelty of their work, particularly with regard to the use of structural equation models to synthesize the available information. I appreciate the inclusion of a paragraph on the limitations of the paper. In turn, they have thoughtfully addressed all of my comments and those of other reviewers. I have only a few relatively minor comments on this version:

Thank you for your thorough review of our revised manuscript titled "Forest carbon stocks increase with higher dominance of ectomycorrhizal trees". We sincerely appreciate your positive feedback and constructive comments.

We are delighted that you found the revised version to be significantly improved and that our efforts to highlight the novelty of our work, especially concerning the application of structural equation models, were effective. Your recognition of the added paragraph addressing the limitations of our study is encouraging, as we aimed to provide a balanced perspective on our findings.

We have carefully considered your comments and have made some adjustments in the manuscript. These changes have been integrated into the revised manuscript, and we believe they enhance the clarity and rigor of our study.

Once again, we appreciate your time and effort in reviewing our manuscript. Your insightful feedback has been instrumental in refining our work, and we are grateful for your continued support in improving the manuscript for publication in Nature Communications.

Please find attached the revised manuscript with track changes highlighting the implemented revisions.

When the authors mention the idea of increasing the dominance of ectomycorrhizal trees, I suggest clarifying that they are referring to native trees -to avoid speculation about planting non-native, sometimes invasive, trees in non-forested areas to sequester C-.

Regarding your suggestion to clarify our statement about increasing the dominance of ectomycorrhizal trees, we agree that it is important to avoid any ambiguity. Specifically, we have revised the manuscript to explicitly specify that our focus is on promoting native ectomycorrhizal trees. This clarification aims to prevent any potential misinterpretation or speculation concerning the introduction of non-native or invasive species in non-forested areas for carbon sequestration purposes.

We have incorporated this clarification into the relevant sections of the manuscript to ensure clarity for the readers (see lines 33 and 302).

I think the hypothesis paragraph should be presented earlier in the text and before the methods paragraph. I suggest that the authors do a better job of connecting their hypothesis to the rest of the introduction.

We have carefully considered your recommendation to present the hypothesis paragraph earlier in the manuscript, preceding the methods section, and to strengthen the connection between our hypothesis and the rest of the introduction. In response, we have reorganized the manuscript to ensure that the hypothesis is introduced prominently at the outset of the introduction, clearly setting the stage for the subsequent discussion.

Specifically, we have moved the hypothesis paragraph to precede the methods section and have revised the introduction to provide a more cohesive narrative that directly links our hypothesis to the broader context of our study objectives and findings. This adjustment aims to enhance the clarity and logical flow of our manuscript, effectively guiding the reader from our research questions to its empirical investigation.

We believe that these revisions significantly improve the manuscript's structure and coherence, aligning more closely with your valuable feedback.

Please find attached the revised manuscript with track changes highlighting the reorganization and improvements made in response to your suggestions (see lines 93-130).

Even though the authors have included a paragraph about the limitations of their work, there are still some sections where they present results as causal when they are actually doing an observational study (e.g. lines 187, 253, 292). I think this can be solved by changing certain words, but I think it is important for the reader to understand the limitations of the implications being discussed.

We acknowledge your concern regarding certain sections where we may have inadvertently presented results as causal despite conducting an observational study. Specifically, you mentioned lines 187, 253, and 292 as examples where greater clarity is needed to convey the observational nature of our findings and to emphasize the limitations of drawing causal conclusions.

To address this issue, we have carefully reviewed these sections and made the necessary revisions. We have adjusted the language to more accurately reflect that our study observes correlations rather than establishes causal relationships. Furthermore, we have reinforced the discussion in these areas to explicitly highlight the limitations of inferring causality from observational data (see lines 188, 253 and 293).

I hope that the above comments can help the authors to further improve this manuscript, which I'm already looking forward to seeing available and citing soon.

We sincerely appreciate your positive evaluation and are grateful for your anticipation of our manuscript's availability. We are committed to addressing your suggestions thoroughly to ensure the manuscript meets the highest standards for publication in Nature Communications.

Thank you once again for your valuable feedback and support. We look forward to the possibility of contributing to the scientific community through this publication and eagerly await your further comments.

Reviewer #6 (Responses in BLUE):

I appreciate the time and care addressing the comments and edits from my review. One comment I had about the climate portion of the hypothesis could still use work on the framing of why climate variability was included in the analyses. As it stands, it still seems like the climate data was just part of this large dataset, so why not include it. It would add to the strength of the paper to see a few sentences or a short paragraph in the intro, not just discussion, on why EcM vs AM associations would interact with differences in climate, beyond just that few studies have looked at this interactive effect. I leave this to the authors' discretion on whether they would like to add that.

Thanks a lot for your thoughtful review and constructive comments on our revised manuscript. We appreciate your careful consideration of our work and your insightful suggestions for improvement.

We acknowledge your comment regarding the framing of the climate portion of our hypothesis and the need to better articulate why climate variability was included in our analyses. To address this suggestion, we have revised the introduction to include a few sentences that explicitly outline the rationale for investigating these interactions (see lines 89-91).

We believe that this clarification will strengthen the manuscript by providing a clearer rationale for the inclusion of climate data in our analyses, thereby enhancing the relevance and impact of our findings.

Thank you once again for your valuable feedback and guidance.